# Cyclin B/CDK1 and Cyclin A/CDK2 phosphorylate DENR to promote mitotic protein translation and faithful cell division

Katharina Clemm von Hohenberg [1,2,3,4], Sandra Müller[1,2], Sibylle Schleich[1,2], Matthias Meister[5], Jonathan Bohlen [1,2,3,6,7], Thomas G. Hofmann [8] & Aurelio A. Teleman [1,2,3✉]

DENR and MCTS1 have been identified as oncogenes in several different tumor entities. The heterodimeric DENR·MCTS1 protein complex promotes translation of mRNAs containing upstream Open Reading Frames (uORFs). We show here that DENR is phosphorylated on Serine 73 by Cyclin B/CDK1 and Cyclin A/CDK2 at the onset of mitosis, and then dephosphorylated as cells exit mitosis. Phosphorylation of Ser73 promotes mitotic stability of DENR protein and prevents its cleavage at Asp26. This leads to enhanced translation of mRNAs involved in mitosis. Indeed, we find that roughly 40% of all mRNAs with elevated translation in mitosis are DENR targets. In the absence of DENR or of Ser73 phosphorylation, cells display elevated levels of aberrant mitoses and cell death. This provides a mechanism how the cell cycle regulates translation of a subset of mitotically relevant mRNAs during mitosis.

[1] German Cancer Research Center (DKFZ), 69120 Heidelberg, Germany. [2] Heidelberg University, 69120 Heidelberg, Germany. [3] CellNetworks—Cluster of Excellence, Heidelberg University, Heidelberg, Germany. [4] Department of Medicine III, Universitätsmedizin Mannheim, 68167 Mannheim, Germany. [5] Division of Viral Transformation Mechanisms, German Cancer Research Center (DKFZ), Heidelberg, Germany. [6] Laboratory of Human Genetics of Infectious Diseases, Necker Branch, INSERM U1163 Paris, France. [7] University of Paris, Imagine Institute, Paris, France. [8] Institute of Toxicology, University Medical Center Mainz at the Johannes Gutenberg University of Mainz, Mainz, Germany. ✉email: a.teleman@dkfz.de

The ability of mammalian cells to adjust their gene expression to their environmental and developmental state is vital[1]. To this end, one of the most important layers of regulation is mRNA translation[2–5]. For instance, in response to stress or quiescence, canonical translation is diminished and cellular translation becomes more dependent on noncanonical factors[6,7]. In proliferating cells, the various phases of the cell cycle likely impose distinct gene expression requirements on the cell, for instance necessitating nucleotide biosynthesis proteins and histones during S-phase or spindle components during mitosis[8,9]. The molecular mechanisms regulating protein translation at different phases of the cell cycle, such as mitosis, are not well understood.

Mitotic translation has been a topic of interest the last few years[10–17]. Several studies have investigated the mitotic regulation of translation, thereby identifying mRNAs that are selectively translated during mitosis[5,8,18–23]. Mitosis is a key feature of proliferative cells, such as tumor cells, hence mitotic translation may represent an Achilles' heel that could be targeted pharmacologically for cancer therapy[24]. Specific molecular mechanisms of translation initiation in mitosis, however, have not been identified.

The DENR·MCTS1 heterodimeric protein complex is involved in noncanonical translation initiation[25,26] and has been linked to cell proliferation and to stress-dependent translation[27–32]. Biochemically, the DENR·MCTS1 complex promotes recycling of post-termination 40S ribosomes[33,34], and its related protein eIF2D is able to recruit initiator tRNA in an eIF2-independent manner on certain viral IRESs[35]. We previously showed that the DENR·MCTS1 complex promotes "translation re-initiation" on mRNAs containing upstream Open Reading Frames (uORFs)[30,36]. Translation reinitiation is the process whereby ribosomes initiate a second round of translation after translating a uORF, rather than dissociating from the mRNA[37,38]. This process is relevant after uORFs with a strong Kozak sequence (stuORFs), which causes them to be translated rather than skipped by leaky scanning. In such cases, re-initiation likely involves stabilization of the post-termination 40S on the mRNA, re-recruitment of initiation factors, resumed scanning, and a new round of initiation on the main downstream ORF. By doing so, DENR·MCTS1 promotes translation of mRNAs involved in neurobiology and in cell proliferation[30–32]. Phenotypically, loss-of-function mutations in DENR are associated with impaired neurocortical migration and brain developmental disorders[39]. Overexpression or copy-number gains of DENR and MCTS1, on the other hand, have been described in several tumor entities[40–42].

In sum, DENR and MCTS1 act in a pro-proliferative and pro-tumorigenic manner and appear to do so by modulating translation of mRNAs containing uORFs. One important open question is whether and how activity of the DENR·MCTS1 protein complex is regulated. MCTS1 can be phosphorylated by Cdc2 on Ser118 and by MAPK on Thr81, the later of the two leading to stabilization of MCTS1 protein[43]. Whether other post-translational modifications affect activity of the complex is not known. We study here regulation of the DENR·MCTS1 complex via phosphorylation. We find that DENR undergoes CDK1- and CDK2-dependent phosphorylation on Ser73 in mitosis to promote translation of specific mitotic target genes that enable timely and faithful cell division and hence mitotic cell survival.

## Results

**DENR is phosphorylated at Serine 73 in mitosis.** To investigate the post-translational regulation of the DENR·MCTS1 complex we undertook three approaches. First, we mutated all sites in the DENR·MCTS1 complex that have been reported on phosphosite.org to be phosphorylated, ubiquitylated or acetylated, and assayed the consequence on DENR·MCTS1 activity using a luciferase translation reporter. Second, we screened all serine/threonine kinases for their ability to phosphorylate DENR·MCTS1 in vitro and further delineated which residues they phosphorylate by mutagenizing DENR and MCTS1. We followed this up by knocking down the kinases in HeLa cells and testing if this affects DENR·MCTS1 activity with the luciferase reporter. All these data are provided to the reader for future reference in the Supplementary Discussion, Supplementary Figs. 8–9, and Supplementary Data 1, 2.

We focus here on the third approach: we aimed to raise phospho-specific antibodies against six phospho-sites that have been detected by mass spectrometry and reported at phosphosite.org (DENR Ser20, Thr69, Ser73, and Ser189 and MCTS1 Thr117 and Ser118). We thereby successfully generated an antibody that specifically detects DENR when phosphorylated on Ser73 (Supplementary Fig. 1a, b). In agreement with this, we analyzed endogenous DENR·MCTS1 immuno-purified from untreated HeLa cells by mass spectrometry and observed phosphorylation of DENR on Ser73 and Thr69 (Supplementary Data 1), confirming that DENR is phosphorylated at Ser73 in vivo. (Generation of phosphoantibodies against the other five sites was not successful, precluding us from studying them further in vivo.) We screened different stresses (ER stress, apoptosis, DNA damage, amino acid and glucose starvation) and environmental conditions (different densities, different cell cycle phases), and discovered that DENR is phosphorylated at Ser73 in a cell cycle-dependent manner (Fig. 1): Synchronization of U2OS cells in S-phase via a double thymidine block or in mitosis via a sequential thymidine-nocodazole treatment, followed by release of each condition revealed that Ser73 phosphorylation is high during mitosis (Fig. 1a, b). Phosphorylation of Ser73 is also elevated in mitotic cells in an unperturbed, asynchronous cell population, as detected by immunostaining (mitotic cells identified by DNA morphology, Fig. 1d). Closer inspection revealed that pDENR(Ser73) is particularly high in early mitotic phases (prophase, prometaphase, metaphase) and then drops (Fig. 1d). (The pDENR staining at the cytokinetic bridge is unspecific as it does not drop upon DENR knockdown, Supplementary Fig. 1c). Also total levels of the DENR·MCTS1 complex vary somewhat throughout the cell cycle, accumulating from G1 to mitosis and decreasing at mitotic exit (Fig. 1a, c). The increase in Ser73 phosphorylation is visible, however, also when normalized to total DENR protein levels (Fig. 1b).

**CDK1/Cyclin B1 and CDK2/Cyclin A2 phosphorylate DENR on Ser73 in mitosis.** We next aimed to identify the kinase responsible for phosphorylating DENR on Ser73 during mitosis. We noticed that Ser73 is positioned within a CDK target consensus motif[44–46] (Supplementary Fig. 2a). We, therefore, tested a panel of CDKs for their ability to phosphorylate DENR Ser73 by in vitro kinase assay and found that both CDK1/Cyclin B1 and CDK2/Cyclin A2 phosphorylate DENR Ser73 in vitro (Fig. 2a, b, Supplementary Data 2). Since both CDK1/Cyclin B1 and CDK2/Cyclin A2 are most active in mitosis or G2/M, these data fit with DENR Ser73 phosphorylation being highest in mitosis (Fig. 1). Interestingly, the differential phosphorylation of DENR·MCTS1 by CDK2 bound to Cyclin A2 (active in G2/M) versus Cyclin E1 (active in G1/S) is a nice example of a cyclin providing substrate specificity to CDK2 (also shown for Cyclin D/CDK4 in[47]). These results fit with the principle that cyclins modulate Cdk specificity, and that as cells progress through the cell cycle towards mitosis Cyclin/CDK complexes become progressively more specific for the consensus CDK phosphorylation motif[45–47].

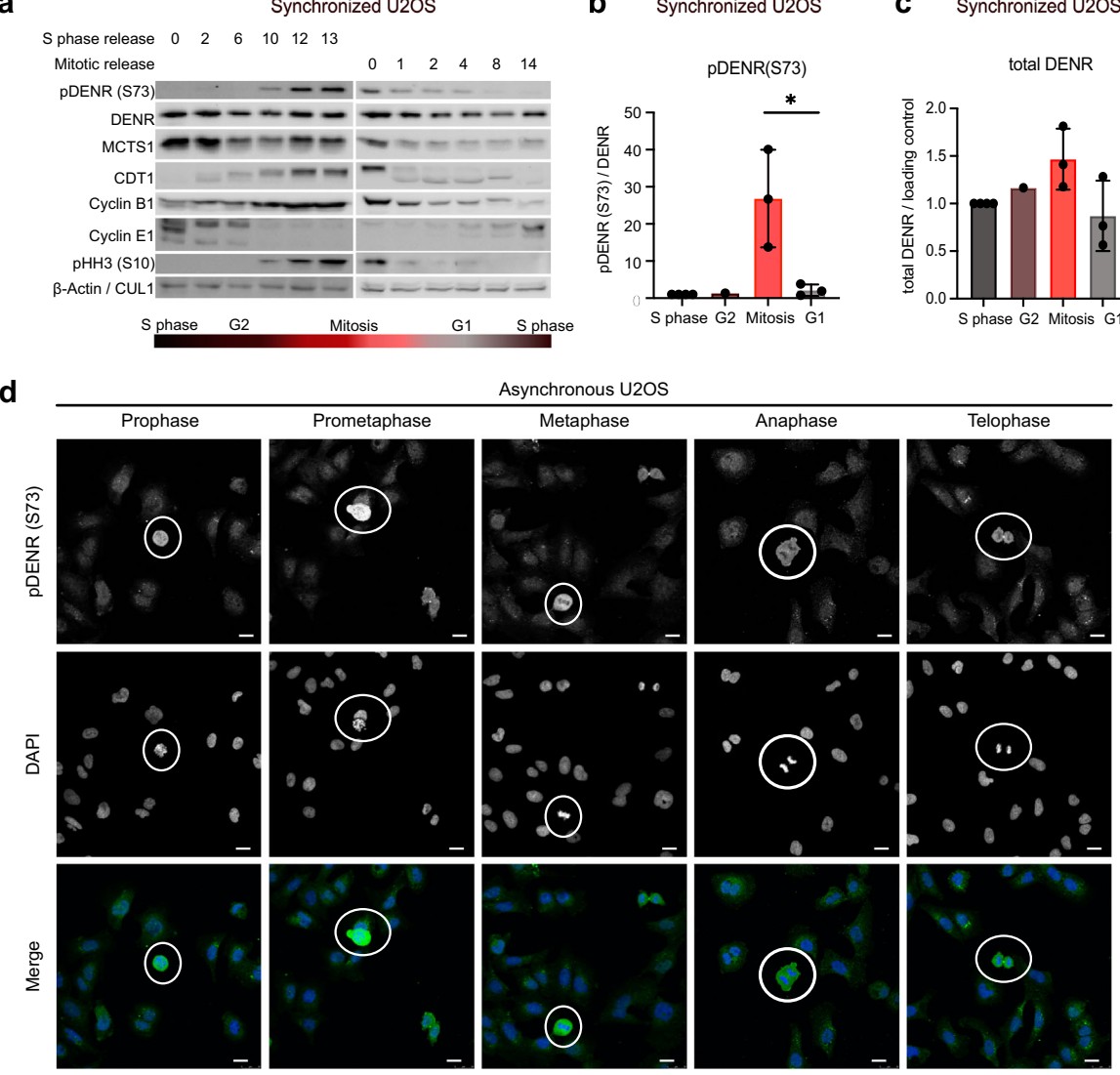

**Fig. 1 DENR protein is phosphorylated at Ser73 during mitosis. a** U2OS cells were synchronized in S phase using a double thymidine block or in mitosis using sequential thymidine and nocodazole treatment, then released, and collected at the indicated time points and subjected to immunoblotting. **b–c** Quantification of Western Blot experiments as described in **a** shows exclusive DENR (S73) phosphorylation and mild accumulation of total DENR protein in mitosis. Means of $n = 1$ (G2, dark-red) or $n = 3$ (S phase, dark-gray, mitosis, light-red, G1, light-gray) independent biological replicates are shown; unpaired $t$-test was performed with *$p$ (mitosis v. G1) $= 0.03$, error bars indicate standard deviations. **d** Asynchronous U2OS cells immunostained using pDENR (S73) antibody show phosphorylation predominantly in early phases of mitosis (prophase, prometaphase, metaphase). Cells in the mitotic phases indicated above the respective images are encircled. Representative images from $n = 3$ independent biological replicates are shown. Scale bars indicate 10 µM.

To test if CDK1 phosphorylates Ser73 in vivo, we synchronized HeLa cells in G2 (lane 2, Fig. 2c), released them to allow entry into mitosis (lane 3) and then added the CDK1 inhibitor RO3306 (lanes 4–6). This revealed a reduction of DENR Ser73 phosphorylation upon CDK1 inhibition (Fig. 2c, d). A similar effect was observed in U2OS cells (Supplementary Fig. 2b).

We next tested if also CDK2 phosphorylates Ser73 in vivo. DENR Ser73 phosphorylation was reduced in mitotic U2OS cells in a dose-dependent manner in response to the CDK2 inhibitors seliciclib or A-674563[48,49] (Fig. 2e, f, Supplementary Fig. 2c), suggesting that CDK2 is also phosphorylating Ser73 in cells. For this experiment, in order to inhibit CDK2/CycA we released cells from a double-thymidine S-phase block, waited for 9 h (Fig. 1a), and then added seliciclib or A-674563. However, to rule out the possibility that the reduction in Ser73 phosphorylation is a secondary consequence of impaired mitotic entry due to reduced CDK1 activation by CDK2/Cyclin A[50], we treated non-synchronized

U2OS cells briefly for one hour with the CDK2 inhibitor seliciclib and then analyzed only the cells that were visibly in mitosis. Also here, DENR phosphorylation was significantly reduced in response to CDK2 inhibition (Supplementary Fig. 2d, e). Consistent with our previous findings, inhibition of DENR Ser73 phosphorylation by seliciclib also reduced DENR protein levels in mitotic cells (Supplementary Fig. 2f), suggesting an effect of this phosphorylation on DENR protein stability. In comparison to U2OS cells, we found that inhibition of CDK2 in mitotic HeLa cells had a less strong effect on Ser73 phosphorylation (Supplementary Fig. 2g). The relative contribution of CDK1 and CDK2 to Ser73 phosphorylation seems to depend on the cell line, with CDK1 predominating in HeLa cells (Fig. 2c, d vs Supplementary Fig. 2g) and CDK2/CycA predominating in U2OS cells (Fig. 2e, f vs Supplementary Fig. 3b). Although seliciclib is commonly used in doses up to 100µM for 24 h and more[48,51], we used only doses up to 10µM for a short time (≤4 h), to assure specificity of this drug. Since seliciclib and

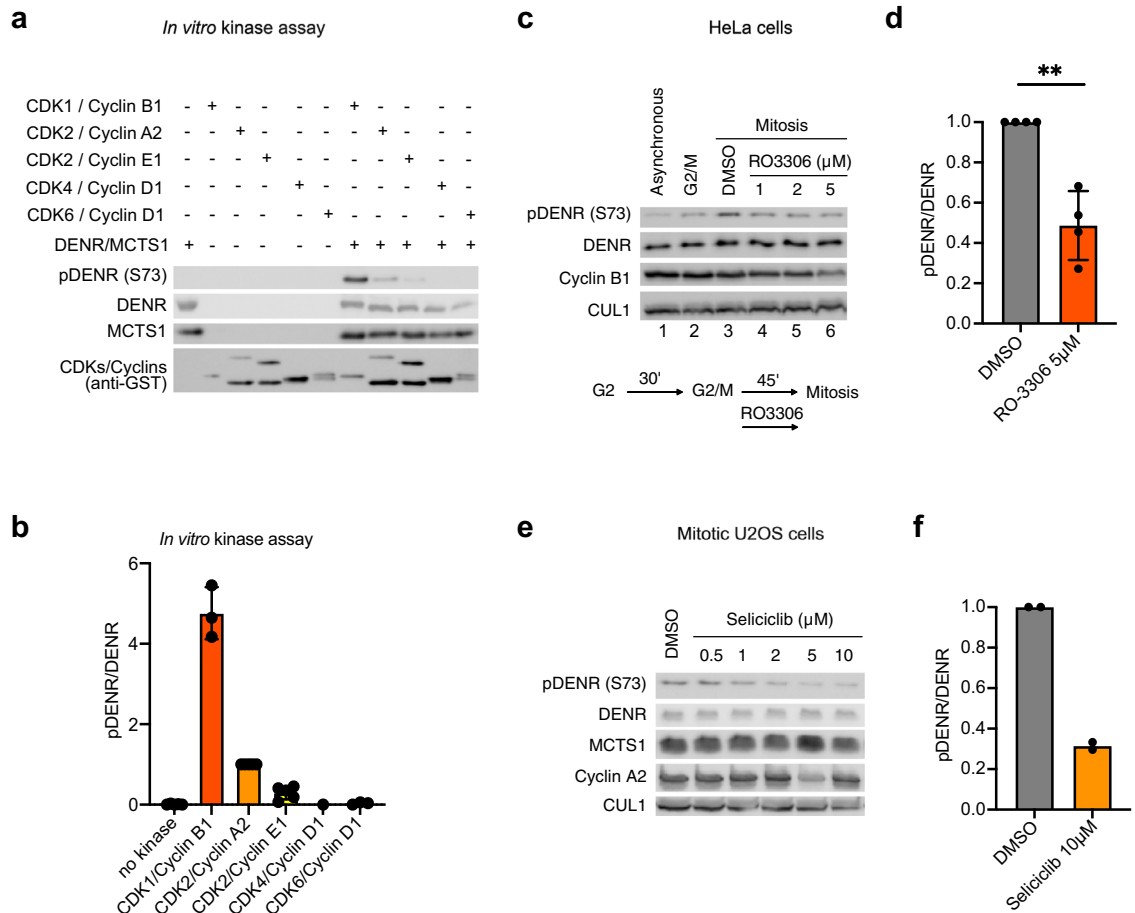

**Fig. 2 Cyclin B/CDK1 and Cyclin A/CDK2 phosphorylate DENR at Ser73 in mitosis. a** In vitro kinase assay using active recombinant Cyclin B1/CDK1, Cyclin A2/CDK2, Cyclin E1/CDK2, Cyclin D1/CDK4 or Cyclin D1/CDK6 (as indicated) on purified DENR·MCTS1 protein complex and analyzed by Western Blot shows phosphorylation of DENR Ser73 by Cyclin B1/CDK1 and Cyclin A2/CDK2. **b** Quantification of biological replicates of the experiment shown in Fig. 2a. Indicated are averages and standard deviations as error bars. $n = 5$ for no kinase and CDK2/CycA (orange), $n = 4$ for CDK2/CycE1 (yellow), $n = 3$ for CDK1/CycB (red) and CDK6/CycD1, $n = 1$ for CDK4/CycD1. **c** Mitotic DENR phosphorylation is reduced upon CDK1 inhibition. HeLa cells were synchronized at end of G2, followed by release and—30 min later—addition of either DMSO or the CDK1 inhibitor RO3306 for 45 min and DENR phosphorylation was assessed by immunoblotting. **d** Quantification of $n = 4$ independent biological replicates of the experiment performed as indicated in **c** with DMSO gray and RO3306 red. Error bars indicate standard deviation and one sample $t$-test was performed with **$p = 0.009$. **e** CDK2 inhibition using seliciclib reduces DENR phosphorylation at Ser73 in U2OS cells that were synchronized in mitosis using a 13 h-release from double thymidine block. DMSO or the indicated doses of seliciclib were added to the cells for the last 4 h. **f** Quantification of $n = 2$ independent biological replicates performed as described in **e** with DMSO gray and seliciclib orange.

A-674563 can inhibit ERK2 activity in vitro at an IC$_{50}$ of 6- to 20-fold the CDK2 IC$_{50}$[48,52], we ruled out that ERK1/2 phosphorylate DENR at Ser73 by using the specific ERK1/2 inhibitor SCH772984[53] (Supplementary Fig. 2h). Together, these in vivo and in vitro data indicate that DENR Ser73 is a direct substrate of CDK1/Cyclin B1 and CDK2/Cyclin A2 in early mitosis.

**Serine 73 phosphorylation protects DENR from mitotic degradation.** To characterize the functional effects of DENR phosphorylation at Ser73, we tested whether a non-phosphorylatable mutant, DENR$^{S73A}$, is impaired in either protein stability or interaction with its binding partner MCTS1[54]. To this end, we expressed FLAG-tagged DENR$^{wildtype}$ or DENR$^{S73A}$ in cells and assayed MCTS1 binding by co-immunoprecipitation. In asynchronous cells, the vast majority of which are in interphase, the levels of FLAG-DENR$^{S73A}$ protein were similar to those of FLAG-DENR$^{WT}$ protein, and both co-immunoprecipitated MCTS1 equally well (Fig. 3a). This is consistent with the fact that Ser73 is not phosphorylated in interphase cells, hence mutation of Ser73 to alanine has little functional consequence in this context. In contrast, in mitotically

synchronized cells we observed dramatically reduced levels of DENR$^{S73A}$ protein compared to wildtype protein, and correspondingly reduced co-immunoprecipitating MCTS1 (Fig. 3b). A cycloheximide time course in mitotically synchronized cells revealed that DENR$^{S73A}$ protein is less stable than wildtype protein (Fig. 3c). Of note, both DENR$^{WT}$ and DENR$^{S73A}$ proteins are stable in interphase cells (cycloheximide time course in Supplementary Fig. 3a) indicating that Ser73 phosphorylation protects DENR from a degradation mechanism that is most active in mitosis. Since we previously showed that DENR and MCTS1 are interdependent on each other for protein stability, and that DENR mutants that cannot bind MCTS1 are unstable[54], this means that the S73A mutation could either directly affect DENR stability, or indirectly affect DENR stability by impairing MCTS1 binding. To distinguish these two options, we aimed to stabilize DENR$^{S73A}$ protein to test its ability to bind MCTS1. We first tested whether proteasomal inhibition with MG132 stabilizes DENR$^{S73A}$ protein, but this was not the case (not shown). Sequence analysis revealed that DENR contains a caspase cleavage site at a.a. 26 (https://web.expasy.org/peptide_cutter/). Although caspases are well known to be active during apoptosis, caspase activity is also involved

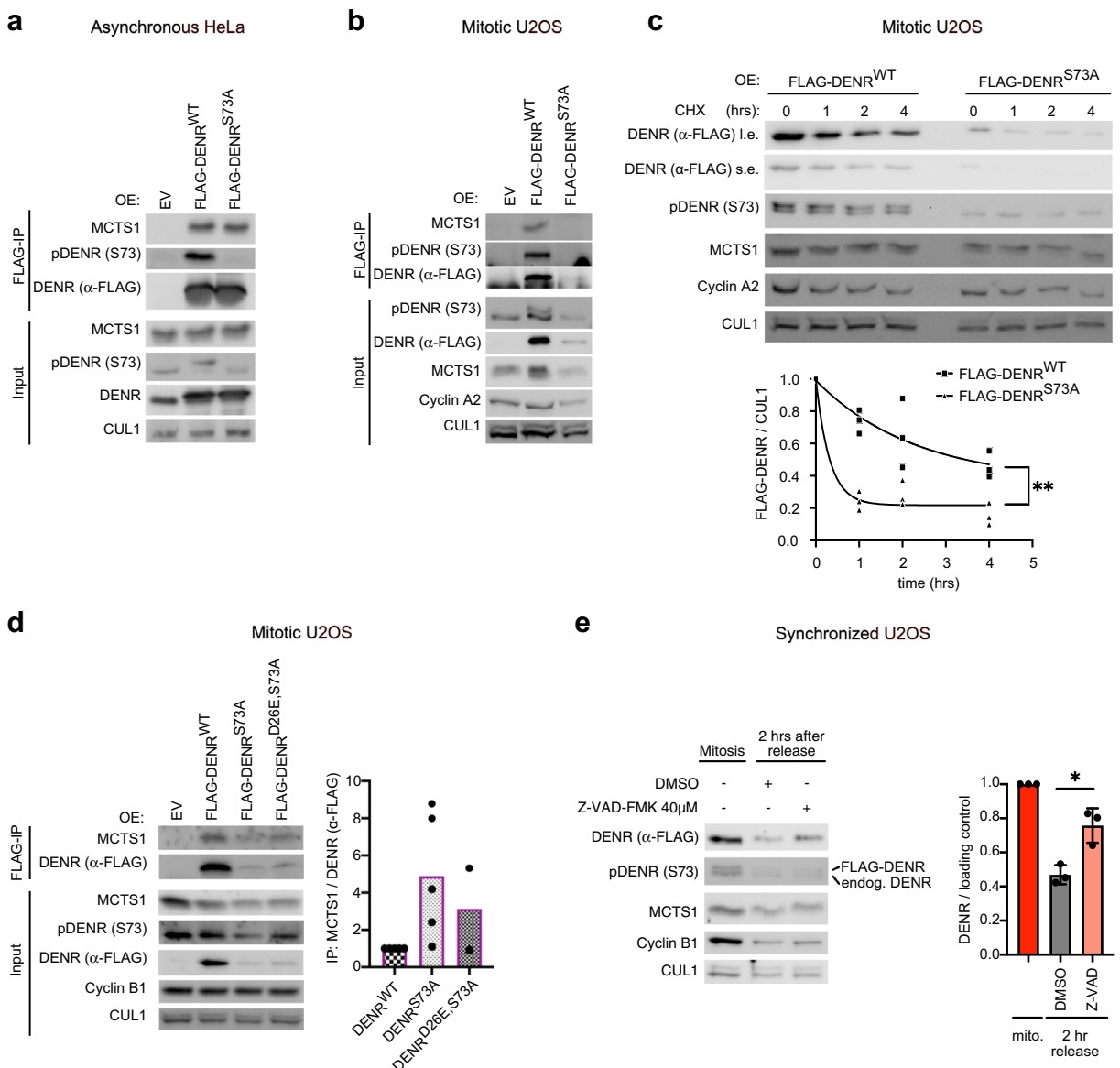

**Fig. 3 Ser73 phosphorylation protects DENR from mitotic degradation. a** Immunoprecipitation of overexpressed FLAG-DENR$^{WT}$ or FLAG-DENR$^{S73A}$ from asynchronous HeLa cells shows MCTS1 binding to both DENR versions. The Western Blot shown is representative of $n = 3$ independent biological replicates. **b** Immunoprecipitation of overexpressed FLAG-DENR$^{WT}$ or FLAG-DENR$^{S73A}$ from mitotically synchronized U2OS cells shows a strong loss of DENR$^{S73A}$ protein. The Western Blot shown is representative of $n = 3$ independent biological replicates. **c** FLAG-DENR$^{S73A}$ is less stable than FLAG-DENR$^{WT}$ in mitotic U2OS cells. Cells were transfected to express either version of DENR, then synchronized in mitosis, harvested by shake-off, and finally exposed to cycloheximide for the indicated times and analyzed by Western Blot. Quantification of FLAG signal normalized to CUL1 from $n = 3$ independent biological replicates. Nonlinear regression lines are shown and Extra sum-of-squares F test of one phase decay constant ($K$) was performed with **$p = 0.009$. **d** Partial restabilization of DENR by the D26E mutation re-establishes binding to MCTS1 in mitotic U2OS cells. This experiment was performed as described in **b** using in addition the non-phosphorylatable, non-cleavable mutant DENR$^{D26E,S73A}$ and STLC instead of nocodazole as a mitotically synchronizing agent. Right panel: quantification of MCTS1 binding to DENR versions. $n = 5$ independent biological replicates for DENR$^{WT}$ and DENR$^{S73A}$ and $n = 2$ for DENR$^{D26E,S73A}$. **e** Destabilization of DENR at mitotic exit is partly caspase-dependent. U2OS cells expressing a FLAG-DENR$^{WT}$ cDNA construct were synchronized in mitosis (red) and then released from mitosis after addition of DMSO (gray) or the pan caspase inhibitor Z-VAD-FMK 40 µM (light-red). Samples were collected at the time of release (before addition of DMSO or Z-VAD-FMK) or 2 h later and analyzed by Western blot. Quantification of total FLAG-DENR signal using ImageJ (v1.50i or 1.53c), normalized to CUL1 and then to the 0 h time point from $n = 3$ independent biological replicates is shown. Error bars indicate standard deviations and unpaired $t$-test was performed with *$p = 0.01$.

in mitosis of normally proliferating cells[55]. Mutation of this putative caspase cleavage site mildly restored the stability of DENR$^{S73A}$ in mitosis (Fig. 3d, Supplementary Fig. 3b) as did the pan-caspase inhibitor Z-VAD-FMK (Supplementary Fig. 3c,d). The fact that stability is not fully rescued suggests that additional cleavage sites or modes of degradation are involved. Nonetheless, the partial recovery of DENR protein stability was accompanied by a corresponding

partial recovery of co-immunoprecipitated MCTS1 (Fig. 3d, Supplementary Fig. 3c) suggesting that Ser73 primarily affects DENR protein stability, not MCTS1 binding. Furthermore, a quantification of the amount of MCTS1 binding per DENR shows that DENR$^{S73A}$ and DENR$^{D26E,S73A}$, if anything, both bind more MCTS1 than DENR$^{WT}$ (Fig. 3d). Consistent with this, DENR binds to MCTS1 during interphase when it is not phosphorylated (Fig. 3a) and

recombinant DENR binds strongly to MCTS1 in bacteria, where Ser73 is not phosphorylated[54].

Ser73 phosphorylation drops when cells exit mitosis (Fig. 1a) starting from anaphase onwards (Fig. 1d), and this drop coincides with a drop in total DENR levels (Fig. 1a, c). We asked if this drop in DENR protein levels during mitotic exit is caspase-mediated. Indeed, the caspase inhibitor Z-VAD-FMK partially rescued the drop in DENR levels as cells exit mitosis (Fig. 3e). In summary, phosphorylation of DENR at Ser73 prevents degradation of DENR during the first stages of mitosis, from prophase to metaphase, while in later mitotic stages and mitotic exit DENR protein is dephosphorylated and cleaved partly in a caspase-dependent manner.

**pDENR (Ser73) induces mitotic translation of DENR target genes.** Since DENR is a translation re-initiation factor, we asked what impact Ser73 phosphorylation has on translation. The data presented above raise the possibility that in mitosis DENR phosphorylation on Ser73 protects it from degradation because DENR is required to drive translation of mitotically relevant mRNAs. We recently performed a RiboSeq analysis of DENR[WT] and DENR[KO] HeLa cells to identify DENR target mRNAs that require DENR for optimal translation[32]. Interestingly, amongst the top DENR targets we found many genes with mitotic functions, primarily concerning the mitotic cytoskeleton (e.g., 10 genes in the top 30, Supplementary Data 3). We first studied translation of these target genes by measuring activity of luciferase reporters carrying the 5′UTRs of these mRNAs (Fig. 4a). The advantage of these luciferase reporters is that they control for transcriptional effects (via the FLuc normalization control reporter) as well as for protein stability effects (via the negative control RLuc reporter). These luciferase assays confirmed that the 5′UTRs of these mitotic genes impart DENR-dependent translation, decreasing in *DENR* knockout cells, and returning to control levels in *DENR* knockouts transfected to re-express DENR (Supplementary Fig. 4a). We tested if translation of these DENR target reporters increases in mitosis, by comparing their activity in asynchronous cells versus cells synchronized in mitosis. This revealed that indeed translation of most reporters increased in mitotic cells by 1.5 to 2-fold, both for DENR targets with mitotic functions (Fig. 4b) as well as DENR targets with no known link to mitosis, which we had previously validated as bona fide DENR targets (Fig. 4c)[32]. Most likely, this assay which is performed on the bulk population of cells underestimates the magnitude of the real mitotic effect due to the fact that available protocols only cause a minor fraction of the cells to synchronize in mitosis. In contrast to reporters carrying the 5'UTRs of endogenous DENR targets, a synthetic reporter carrying a stuORF did not increase translation in mitosis (Fig. 4b), suggesting it lacks an element required for mitotic translation. The increased translation of these reporters in mitosis is due to elevated mitotic DENR activity because it is blunted in *DENR* knockdown cells (Fig. 4d–g, Supplementary Fig. 4b–d), and it is reversed by CDK2 inhibition with seliciclib (Supplementary Fig. 4e). The mitotic induction of the DUSP4 and CDKL5 reporters was abolished when the ATGs of the uORFs were mutated (Supplementary Fig. 4f), consistent with DENR·MCTS1 promoting translation re-initiation after uORFs[32]. In sum, these data indicate that DENR activity increases in mitosis compared to interphase.

We previously identified DENR targets by performing RiboSeq on asynchronous HeLa cells[32]. To test whether DENR promotes translation of the same set of target mRNAs in mitotic cells as in interphase cells, or whether phosphorylation on Ser73 might affect the DENR target set, we performed RiboSeq and RNA-seq on mitotic and interphase DENR[WT] and DENR[KO] cells

(Supplementary Data 4). Agents used to synchronize cells in mitosis such as nocodazole, and to a lesser extent the Eg5 inhibitor STLC, induce cell stress and hence perturb the translatome[56,57]. We therefore used alternative methods to enrich or deplete mitotic cells from our populations. For the mitotic population, we synchronized HeLa cells using a double thymidine block and collected them for analysis 9 h after the second S phase release. For the interphase sample, we shook off and discarded mitotic cells, which anyways constitute a small minority (<5%) of an asynchronous population. A Z-vs-Z analysis of the RNA-seq data from wildtype cells identified 1326 transcripts corresponding to 694 genes with elevated mRNA levels in mitosis (Supplementary Fig. 5a, Supplementary Data 5) and none with reduced mRNA levels, probably due to the short duration of mitosis. A comparison of translation efficiency (ribosome footprints normalized to mRNA) between wildtype mitotic and interphase cells identified 266 transcripts (181 genes) that were translationally up-regulated in mitotic cells and 1090 transcripts (696 genes) that were translationally down-regulated in mitotic cells (Supplementary Fig. 5b, Supplementary Data 6). A comparison of translation efficiency in DENR[KO] versus DENR[WT] cells identified 1108 transcripts (653 genes) and 990 transcripts (576 genes) as DENR targets in interphase and mitotic cells, respectively (Supplementary Fig. 5c, d, Supplementary Data 7). Interestingly, a comparison of the change in translation efficiency upon *DENR* loss in interphase versus mitotic cells showed a good correlation (Supplementary Fig. 5e) suggesting that in general the mRNAs that are DENR targets in mitosis are also DENR targets in interphase, although to varying degrees. A hand-full of mRNAs appear to be strong DENR targets in interphase cells but not in mitotic cells (Supplementary Fig. 5e), but these genes are amongst the ones with the strongest drop in translation efficiency transcriptome-wide in wildtype mitotic cells compared to interphase cells (e.g., IL11). This suggests that these mRNAs are not well translated in mitosis, and hence are not sensitive to DENR loss. In sum, since the set of DENR target mRNAs does not change during mitosis, this suggests that phosphorylation of DENR on Ser73 in mitosis affects its stability but not another aspect of its function. Interestingly, of the 266 transcripts that are translationally up-regulated during mitosis in wildtype cells, 114 are DENR targets. This fraction (~40%) is significantly higher than the 10% of transcripts that are DENR targets in interphase cells ($p = 0.0$ by binomial distribution). Hence DENR appears to play a particularly important role in mitotic translation.

To investigate if DENR-dependent translation affects the level of its targets during mitosis, we first performed Western Blot analysis of asynchronous and mitotic cells and found that the level of DENR target proteins increases in mitosis (Supplementary Fig. 6a). Interestingly, we observed that the mobility of DENR protein on the SDS-PAGE gel shifts completely upwards to a slower migrating form in mitotically synchronized cells (Supplementary Fig. 6a), suggesting that DENR is highly phosphorylated in mitotic cells. We next asked whether this increase in protein levels is DENR dependent. To this end, we immunostained unsynchronized DENR[WT] or DENR[KO] cells for target proteins and quantified protein levels specifically in mitotic cells identified by chromosome morphology. This revealed a significant decrease in target protein levels upon loss of DENR in mitotic cells (Fig. 4h, i, Supplementary Fig. 6b, c). The same could be observed by western blotting lysates of mitotically synchronized cells (Supplementary Fig. 6d). A drop in target protein levels was also present in asynchronous cells, although less dramatic than in mitotic cells (Supplementary Fig. 6e). These drops in target protein levels were rescued by reconstituting the DENR[KO] cells with a DENR[WT] expression construct (Supplementary Fig. 6d), confirming they are on-target effects. When,

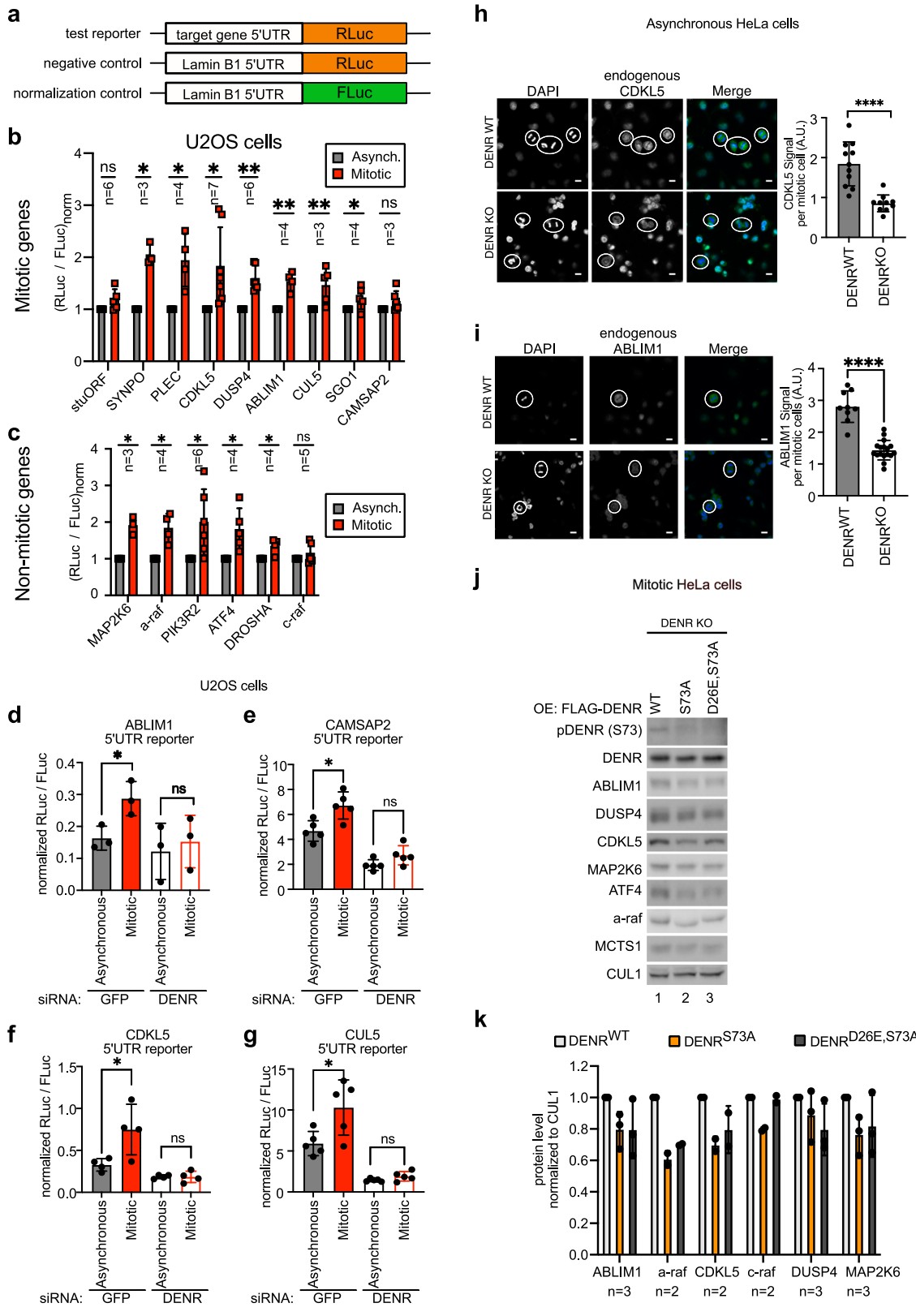

however, reconstitution was performed with the non-phosphorylatable DENR$^{S73A}$ mutant, rescue of target protein levels in mitotic cells was impaired (compare lane 2 to lane 1 in Fig. 4j, k, Supplementary Fig. 6f). Since DENR is phosphorylated at Ser73 only in mitotic cells, this confirms there is DENR-dependent translation ongoing during mitosis. In accordance

with our previous findings showing that mutation of the caspase cleavage site partially reconstitutes DENR stability, mitotic target protein levels are also partially rescued when the non-phosphorylatable, non-cleavable version of DENR (DENR$^{D26E,S73A}$) is expressed in DENR$^{KO}$ cells (Fig. 4j, k). In sum, our findings show that there is a set of mitotically relevant mRNAs, such as CDKL5

**Fig. 4 pDENR (S73) induces mitotic translation of DENR target genes. a** Reporter constructs used in this figure. A LaminB1 5'UTR-firefly luciferase (FLuc) reporter was used as a normalization control in all wells. Luminescence of target gene 5'UTR-renilla luciferase (RLuc) reporters was normalized to an equivalent negative control reporter carrying the LaminB1 5'UTR. **b, c** Luciferase reporters carrying the 5'UTRs of the indicated DENR target genes exhibit increased translation in mitotic U2OS cells (sequential thymidine-nocodazole (red)) compared to those left unsynchronized (gray). One sample $t$-tests on $n$ independent biological replicates: stuORF $p = 0.4$, SYNPO *$p = 0.01$, PLEC *$p = 0.03$, CDKL5 *$p = 0.02$, DUSP4 **$p = 0.002$, ABLIM1 **$p = 0.01$, CUL5 **$p = 0.01$, SGO1 *$p = 0.03$, CAMSAP2 $p = 0.1$, MAP2K6 *$p = 0.02$, a-raf *$p = 0.02$, PIK3R2 *$p = 0.04$, ATF4 *$p = 0.03$, DROSHA *$p = 0.045$, c-raf $p = 0.3$. **d–g** Translation of reporter constructs carrying 5'UTRs of DENR target genes increases in mitosis in a DENR-dependent manner. GFP- (filled) or DENR- (empty) siRNA-treated U2OS cells were transfected with the indicated reporters, and synchronized in mitosis (red) or not synchronized (gray). Means and unpaired $t$-test from n biological replicates. **d** $n = 3$, *$p = 0.03$, ns $p = 0.68$, **e** $n = 5$, *$p = 0.01$ ns $p = 0.08$, **f** $n = 4$, *$p = 0.04$, ns $p = 0.96$, **g** $n = 5$, *$p = 0.03$, ns $p = 0.12$. Endogenous levels of DENR target proteins CDKL5 (**h**) or ABLIM1 (**i**) are reduced in mitotic DENR knockout versus mitotic control cells. Unsynchronized HeLa DENR[WT] (gray) or DENR[KO] (white) cells immunostained for CDKL5 or ABLIM1. For quantifications, **h** $n = 11$ DENR[WT] and $n = 10$ DENR[KO] mitotic cells. **i** $n = 9$ DENR[WT] and $n = 17$ DENR[KO] mitotic cells. Means of signal intensities are displayed. **** unpaired $t$-test $p < 0.0001$. Scale bars indicate 10 μM. **j** Reconstitution of DENR[KO] cells with DENR[WT] rescues mitotic protein levels of target genes, but not reconstitution with DENR[S73A] and only partially with DENR[D26E,S73A]. HeLa DENR[KO] cells were transfected with plasmids carrying the indicated DENR versions and a puromycin resistance gene, selected with puromycin, synchronized (sequential thymidine-STLC) and collected for analysis by Western Blot. **k** Quantification of protein levels for independent biological replicates of the experiment shown in panel j with DENR[WT] light-gray, DENR[S73A] orange and DENR[S73A,D26E] dark-gray. In all subfigures, error bars indicate standard deviations.

mRNA, that are translated in a DENR-dependent manner in mitosis.

**pDENR (Ser73) prevents aberrant mitosis and promotes faithful cell division.** Since the DENR·MCTS1 complex promotes translation of mRNAs with mitotic functions, we asked if DENR·MCTS1 is required for proper mitosis. To this end, we performed two-dimensional flow cytometry of unsynchronized DENR[WT] or DENR[KO] cells and observed a four-fold accumulation of DENR[KO] cells in mitosis (Fig. 5a, Supplementary Fig. 7a), raising the possibility of a defect in progression through mitosis. Indeed, an elevated number of mitotic DENR[KO] cells undergo apoptosis (Fig. 5b, Supplementary Fig. 7a). This suggests that loss of DENR leads to slower mitotic progression and increased mitotic failure, likely contributing to the reduced proliferation rate of DENR[KO] cells which we previously reported[32]. We then examined whether there are any mitotic defects in DENR[KO] cells (Fig. 5c, d). While the fraction of early mitotic phases (prophase, prometaphase, metaphase) is not significantly influenced by the absence of DENR, there is a reduction in the number of cells in the late mitotic phases (anaphase, telophase), and instead an accumulation of atypical mitotic figures and mitotic blebs, representative of mitotic cell death (Fig. 5c, d). Taken together with the increase in the fraction of mitotic cells, these findings indicate that the early phases of mitosis are prolonged in DENR[KO] cells, and in some DENR[KO] cells mitotic cell death occurs after anaphase onset. This effect can be rescued by re-expression of DENR[WT] in DENR[KO] cells, but not by the non-phosphorylatable DENR[S73A] mutant, and only partially by the non-phosphorylatable, non-cleavable DENR[D26E,S73A] mutant (Fig. 5c, d). We observed a similar phenotype in asynchronous *DENR*-knockdown U2OS cells (Supplementary Fig. 7b), ruling out a cell type-specific defect in HeLa cells. Interestingly, we observed an increased fraction of irregular and multipolar spindles in DENR[KO] compared to DENR[WT] cells (Supplementary Fig. 7c), which could explain in part the mitotic failure in DENR-depleted cells. Since DENR[KO] cells reconstituted with DENR[S73A] have mitotic defects, and since Ser73 is specifically phosphorylated in mitosis, these data show that the translation occurring during mitosis is important for mitosis to proceed correctly. Furthermore, the DENR/MCTS1 complex appears to be playing a particularly important role during mitosis because 114 of the 226 transcripts which are translationally upregulated during mitosis are DENR targets (i.e., 40%), which is a significantly larger fraction than the 10% of transcripts that are DENR targets in interphase cells ($p = 0$ by binomial distribution).

In sum, phosphorylation of DENR at Ser73 and stabilization of the DENR·MCTS1 complex in the early phases of mitosis is important to guarantee efficient cell division and to prevent aberrant mitosis. It is likely that the mitotic defects that result from loss of DENR or DENR Ser73 phosphorylation (Fig. 5d) reflect a combined contribution of multiple DENR targets, since multiple DENR target mRNAs are insufficiently translated during mitosis in the absence of DENR activity (Fig. 4, Supplementary Fig. 6), and multiple DENR targets have mitotic functions including spindle dynamics (Supplementary Data 3).

## Discussion

We have identified a signaling pathway promoting mitotic translation of genes that are crucial for faithful and timely cell division (Fig. 6). At the onset of mitosis CDK1/Cyclin B and CDK2/Cyclin A phosphorylate the non-canonical translation initiation factor DENR at serine 73 and thereby protect DENR from degradation (Figs. 1–3). DENR then acts to promote translation of a set of target genes that are known to be involved in proper cell division (Fig. 4), thereby supporting mitotic progression and preventing aberrant mitosis and mitotic cell death (Figs. 5, 6). Interestingly, almost half of all mRNAs with increased translation in mitosis depend on DENR for their translation, suggesting that DENR activity is particularly important during this phase of the cell cycle.

Previous studies have identified mRNAs whose translation is upregulated during mitosis, however, the functional significance of this up-regulation for mitotic progression was difficult to test, given that there was no intervention known to specifically block mitotic translation. Since DENR phosphorylation on Ser73 is specific for mitosis, this enables such an intervention. The fact that reconstitution of DENR[KO] cells with DENR[S73A] does not rescue their mitotic defects indicates that the translation occurring during mitosis is indeed important for mitosis itself.

This regulatory mechanism is initiated through DENR phosphorylation by CDK1 and CDK2. Interestingly, we find that Cyclin A plays an important role in enabling CDK2 to phosphorylate DENR, likely via substrate recognition, because CDK2 bound to Cyclin A phosphorylates DENR in vitro more efficiently than CDK2 bound to Cyclin E (Fig. 2a, b). This is consistent with the fact that in vivo DENR Ser73 is phosphorylated in mitosis, when CycA/CDK2 is active, but not during S-phase, when CycE/CDK2 is active (Fig. 1a).

CDK2 and CDK1 are vital in tumors and therefore attractive targets for cancer therapy. Different chemical CDK2 inhibitors have been developed and the most advanced one, seliciclib, is

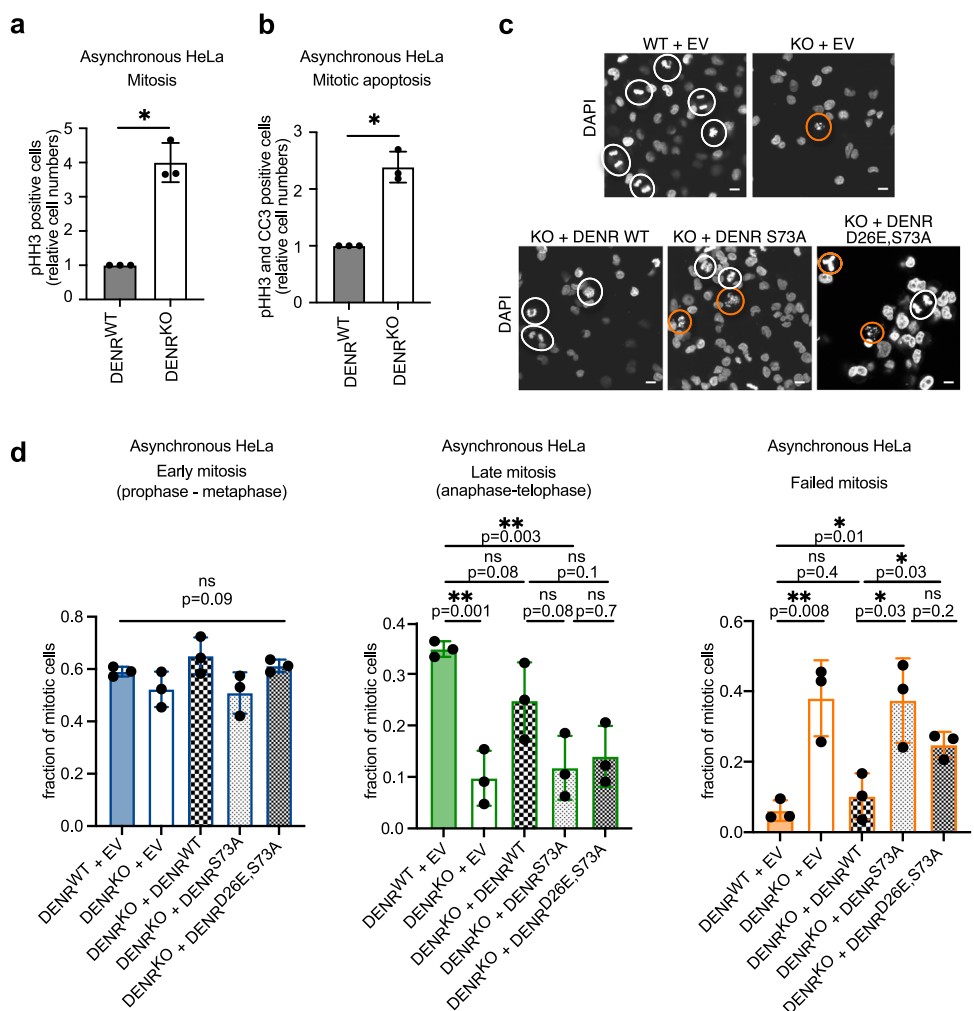

**Fig. 5 pDENR (S73) prevents aberrant mitosis and promotes faithful cell division.** DENR knockout cells have an elevated proportion of cells in mitosis (**a**) that are apoptotic (**b**). Unsynchronized HeLa DENR[WT] (gray) or DENR[KO] (white) cells were fixed, permeabilized, stained for cleaved caspase 3 (CC3) and pHH3 (S10) and then analyzed by flow cytometry. Shown are means, normalized to DENR[WT], and standard deviations as error bars from $n = 3$ biological replicates. One sample $t$-test with $*p = 0.01$ for both **a** and **b**. DENR knockout cells have a reduced proportion of cells in late mitotic phases and elevated aberrant mitoses and mitotic cell death, both of which are rescued by reconstitution with DENR[WT] (checkered) but not with DENR[S73A] (finely checkered light) and only partially with DENR[D26E,S73A] (finely checkered dark) protein. Unsynchronized HeLa DENR[WT] (filled) or DENR[KO] (empty) cells were transfected with empty expression vector (EV) or one of the indicated FLAG-tagged constructs for 72 h, then fixed and stained with DAPI. **c** Sample images. Mitotic cells are outlined by white circles. Examples of failed mitoses (multipolar spindles, mitotic blebs/mitotic cell death) are encircled in orange. Scale bars indicate 10 μM. **d** Quantification of mitotic phases and mitotic defects, assessed via DAPI stain. Displayed are means of fractions of all mitotic cells, and standard deviations as error bars from $n = 3$ independent biological replicates. Early mitosis comprises prophase, prometaphase and metaphase, late mitosis comprises anaphase and telophase and failed mitosis comprises aberrant mitosis or mitotic blebs. Two-way ANOVA was performed for Early mitosis (blue) and unpaired $t$-tests were performed for Late mitosis (green) and Failed mitosis (orange).

currently being tested in clinical studies. While its molecular action has only been partially discovered, seliciclib is known to prevent faithful cell division and cause cell death in mitosis, due to a type of uncoordinated cellular division called anaphase catastrophe[58,59]. From the fact that DENR loss of function phenocopies this mitotic failure (Fig. 5, Supplementary Fig. 7), it is possible that part of the effect of seliciclib might be due to inhibition of DENR phosphorylation at Ser73 and accordingly impaired translation of DENR-dependent mitotic target genes that act towards coordinated cell division and cytokinesis.

In line with our observations that DENR·MCTS1 protein complex is essential for faithful cell division, mitotic catastrophe and delayed cytokinesis have been observed in cells depleted of MCTS1[60], however, the mechanism of this MCTS1 effect was unknown. Since DENR and MCTS1 are co-dependent on each other for protein stability and act together as one functional

heterodimeric complex, it is likely that reduced translation of the mitotic target genes as we describe here is one contributing factor.

We observed that DENR is stabilized by phosphorylation at serine 73 in mitosis. When not phosphorylated, DENR is degraded in a manner that is partly caspase-dependent (Fig. 3d, e, Supplementary Fig. 3). Although caspases are well known to be active during apoptosis, there is also increasing evidence of non-apoptotic, cell cycle-related functions for caspases[61]. Caspases 2 and 7 have been described to drive mitotic exit[55,62–64]. Consistent with this, we observed caspase-dependent degradation of DENR during the late phases of mitosis (anaphase, telophase). Interphase DENR, however, is very stable independently of its phosphorylation status (Fig. 3a, Supplementary Fig. 3a). This suggests that caspase activity towards DENR increases in mitosis. We noticed that caspases 2 and 3 are described as being nuclear (https://www.proteinatlas.org/ENSG00000106144-CASP2/cell,

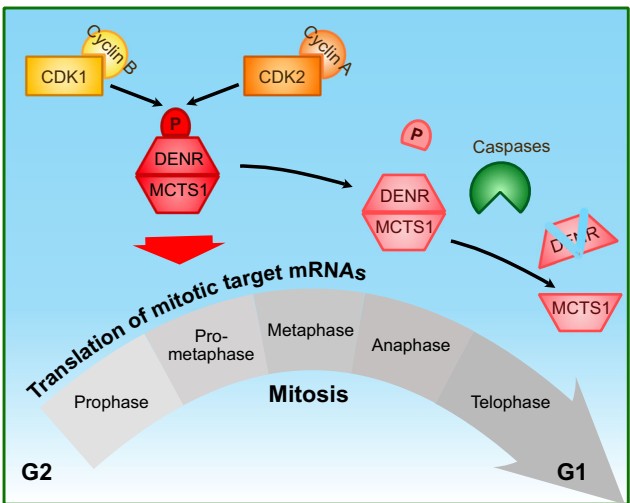

**Fig. 6 Schematic summary of DENR regulation in mitosis.** During early stages of mitosis, DENR protein is stabilized via phosphorylation of Ser73 which enables it to promote translation of target genes involved in mitosis. As of anaphase, DENR phosphorylation on Ser73 decreases, concomitant with a caspase-dependent decrease in DENR protein levels as cells exit mitosis.

https://www.proteinatlas.org/ENSG00000164305-CASP3/cell), so one could speculate that, breakdown of the nuclear envelope renders DENR accessible to otherwise nuclear caspases and therefore allows for a surge of DENR cleavage unless prevented by Ser73 phosphorylation. This interplay could be topic of future study. That said, the rescue of DENR stability either upon mutating the caspase site in DENR or with the Z-VAD-FMK caspase inhibitor is only partial (Fig. 3d, e), suggesting additional degradation mechanisms may be at play.

We noticed that unlike other reporter constructs carrying the 5′UTRs of endogenous DENR target mRNAs, our synthetic stuORF reporter does not increase in translation during mitosis (Fig. 4b). One possible explanation we explored is that during mitosis DENR Ser73 phosphorylation changes DENR function so that it no longer acts on one set of interphase target mRNAs (represented by the stuORF reporter) and instead it acts on a distinct set of mRNAs. From our ribosome profiling experiment, however, this does not seem to be the case (Supplementary Fig. 5e). There is a good correlation between the mRNAs that are DENR targets in interphase and in mitosis, suggesting there are not two distinct sets of target genes. The few target genes that are strongly DENR dependent in interphase but less so in mitosis (e.g., IL11, ELFN2, and FBXO46) are poorly translated in mitosis. The IL11 mRNA for instance is the most down-translated mRNA transcriptome-wide in wildtype mitotic cells compared to inter-phase cells (Supplementary Data 6). Hence if these mRNAs are not translated in mitosis, the presence or absence of DENR cannot affect them. Therefore, although we do not know why our synthetic stuORF reporter does not increase its translation during mitosis, we believe the most likely explanation is that it is not being translated in mitosis for some technical reason.

We present here in Supplementary Materials a few unbiased and comprehensive screens for pathways that might regulate the DENR·MCTS1 complex. We tested all serine/threonine kinases by in vitro kinase assay, we knocked down all hits from this kinase screen and assayed DENR·MCTS1 activity using the stuORF reporter in HeLa and MCF7 cells, and we systematically mutated all amino acids in DENR or MCTS1 which were

reported in public databases to be post-translationally modified. While we focused here on the CDK axis, our data suggest that there might be additional regulatory pathways that could be further explored. One example is the in vitro phosphorylation of MCTS1 at threonines 81 and 179 by STK3 (Supplementary Fig. 9d). Additional examples are the effects of *PRKCQ* or *PRKG1* knockdown on DENR·MCTS1/stuORF activity (Supplementary Fig. 9a, b). Furthermore, abrogation of possible posttranslational modifications on MCTS1 lysine 51 showed a mild but significant reduction in stuORF activity (Supplementary Fig. 8e), suggesting acetylation or ubiquitylation might provide additional levels of regulation to this translation complex. These might be interesting starting points for future studies on the regulation of the DENR·MCTS1 complex.

## Methods

**Cloning**. Sequences of oligos used for cloning are provided in Supplementary Data 4. For expression of MCTS1 or DENR and their mutants (Supplementary Fig. 8b, c) the human MCTS1 or DENR coding sequences were cloned into a pRK vector backbone (pAT1063, Teleman lab collection) via restriction sites EcoRI and ClaI. Then four silent mutations within exon 6 were inserted into the *MCTS1* coding sequence by site-directed mutagenesis to make it resistant against MCTS1 siRNA. Likewise, all mutations of the *DENR* and *MCTS1* coding sequence were inserted by site-directed mutagenesis (Supplementary Fig. 8b, c, Supplementary Data 1). For negative controls, the empty pRK vector backbone was used. In order to clone FLAG-tagged DENR constructs (Fig. 3, Fig. 4j, k, Fig. 5c, d, Supplementary Figs. 1b, 3, 6f, 8b–e), DENR (wildtype, S73A or D26E,S73A mutant) coding sequence was amplified using primers containing a C-terminal FLAG-tag and again cloned into the above mentioned pRK vector backbone using EcoRI and ClaI restriction sites or into a pcDNA3.1 vector containing a puromycin resistance (pGF045) using BamHI and EcoRI restriction sites.

In order to obtain mutated constructs for in vitro kinase assays (Supplementary Fig. 9c, d) site-directed mutagenesis was performed on a plasmid (pET-DUET-1, pSS290)[54] coding for both His-tagged DENR^WT and His-tagged MCTS1^WT, before recloning into the pET-DUET-1 vector using XbaI and NotI (DENR) or MunI/EcoRI and XhoI (MCTS1) as restriction sites.

The Lamin B1 5′UTR firefly and renilla luciferase reporters and the Lamin B1 5′ UTR stuORF reporter were obtained from a previous project in our lab[30]. Renilla luciferase reporters with 5′UTRs of various genes (Fig. 4b–g, Supplementary Fig. 4) were cloned by amplifying the 5′UTR of the gene of interest from cDNA and cloning it into the renilla luciferase reporter plasmid at the HindIII and Bsp119l sites. Cloning of the reporter plasmids for ATF4, a-raf, c-raf, DROSHA, MAP2K6, and PIK3R2 was performed likewise[32].

All sequences of primers used for cloning are detailed in Supplementary Data 8.

**Expression and purification of the DENR·MCTS1 protein complex**. Proteins were expressed using *E. coli* BL21 (DE3) cells in 2YT media supplemented with Kanamycin (30 μg/ml). Cells were grown to an $OD_{600}$ of 0.8–1.0 at 37 °C, then shifted to 18 °C. Expression was induced with the addition of 0.4 mM IPTG, and cells were grown further overnight, harvested by centrifugation, and the cell pellets either used immediately for lysis and purification or frozen with $LN_2$ and stored at −20 °C.

All variants of the DENR·MCTS1 complex (Supplementary Fig. 9c, d, Supplementary Data 2) were purified via a C-terminal His_6-tag using NiNTA and size exclusion chromatography. Cells were resuspended in lysis buffer (30 mM HEPES, 30 mM Imidazol, 500 mM NaCl) and lysed with a Microfluidizer (Microfluidics) at 0.55 MPa. The lysate was cleared by centrifugation for 35 min at 35,000 × g and 4 °C, and the resulting supernatant was applied to a 2 ml NiNTA column. The column was washed with 25–50 column volumes of lysis buffer and eluted with elution buffer (lysis buffer plus 400 mM Imidazol). The NiNTA-eluate was applied to a Superdex 200 26/60 column, equilibrated with SEC-buffer I (10 mM HEPES pH 7.5, 500 mM NaCl). Peak fractions containing the DENR–MCTS1 complex were pooled, concentrated to 10–15 mg/ml, and either used directly or shock-frozen with $LN_2$ and stored at −80 °C.

**Cell culture**. HeLa cells (American Type Culture Collection, ATCC #CCL-2), MCF7 cells (ATCC #HTB-22) and U-2 OS cells (ATCC #HTB-96) were cultured in Dulbecco's modified Eagle's medium (DMEM), supplemented with 10% fetal bovine serum and 1% penicillin/streptomycin. HeLa DENR^KO cells were generated in our lab[32]. For this manuscript clone no. 3.42 was used throughout, only for RiboSeq analysis clone 2.11 was added. All cell lines were tested negative for mycoplasma contamination.

The concentration of all drugs used for synchronization of cells are specified in the "drug treatments" section below. For synchronization of U2OS cells in mitosis

(Figs. 1a–c, 3b–h, 4b–g, Supplementary Figs. 3c, d, 4b–d, f) cells were plated in thymidine. 24–28 h later medium was removed and cells were washed twice with PBS and once for 5 min with normal medium. Then, fresh medium containing nocodazole or STLC was added for 13 h. For mitotic synchronization, HeLa cells (Fig. 4j, k, Supplementary Fig. 6a, f) were released from S phase arrest (thymidine) for 9 h (Fig. 4j, k, Supplementary Fig. 6f) or 13 h (Supplementary Fig. 6a) in the presence of STLC, followed by Western Blot analyses or luciferase assay. For mitotic enrichment (Supplementary Fig. 6d) HeLa cells were released from a single thymidine block into normal medium for 9 h and then analyzed as indicated.

For mitotic release (Figs. 1a–c, 3g, h) nocodazole-arrested U2OS cells were shaken off, spun down, washed twice in PBS and once in medium and then replated in normal medium. The first sample ("mitotic release 0") was collected at the time of mitotic shake-off.

For assessment of mitotic stability (Fig. 3c, d, Supplementary Fig. 3b) U2OS cells were synchronized with thymidine and nocodazole as described above, then they were shaken off, spun down, and collected or replated in medium containing nocodazole and cycloheximide at a concentration of 100 µg/ml.

For analysis of phosphorylation during mitotic entry U2OS (Fig. 2e, f, Supplementary Fig. 2c) or HeLa (Supplementary Fig. 2g) cells were plated in thymidine. 24–28 h later medium was removed and cells were washed twice with PBS and once for 5 min with normal medium and then fresh medium was added. In U2OS cells, 10 h later thymidine was added again. 24–28 h later cells were released again (washed twice in PBS, once in medium) into nocodazole-containing medium. 9 h later DMSO or the indicated inhibitor was added and 4 h later cells were harvested and analyzed by Western blot or luciferase assay. HeLa cells were released from first thymidine arrest immediately into STLC-containing medium. 6 h after release DMSO or the indicated inhibitor was added and three hours later cells were harvested and analyzed by Western blot.

For synchronization in G2 phase U2OS (Supplementary Fig. 2b) or HeLa (Fig. 2c, d) were plated in thymidine-containing medium for 24–28 h, then released into medium containing the CDK1 inhibitor RO3306 at a concentration of 5 µM. 13 h (U2OS) or 9 h (HeLa) later – "G2" –, cells were released again using PBS and medium for 20–30 min – "G2/M" –, and then DMSO or RO3306 in the indicated concentrations was added for another 45–60 min – "Mitosis" –, followed by collection and analysis by Western Blot.

For S phase release (Fig. 1a–c) U2OS cells were subjected to a double thymidine block as described above. After the second thymidine block cells were released, replated in nocodazole-containing medium and collected at the indicated time points.

**Drug treatments**. Where indicated, and if not otherwise specified, the following drugs were used: Z-VAD-FMK 40 µM (SelleckChem, #S7023), thymidine 2 mM (Sigma, #T1895-1G), nocodazole 400 ng/ml (Sigma, #SML1665-1ML), cyclohex-imide 100 µg/ml (Santa Cruz, #sc-3508), RO3306 5 µM unless otherwise specified (Sigma #SLM0569-5MG), seliciclib 10 µM unless otherwise specified (SelleckChem, #S1153), A-674563 1 µM unless otherwise specified (SelleckChem, #S2670), STLC 5 µM (Sigma, #164739-5 G). Z-VAD-FMK was added at the time of FLAG-DENR plasmid transfection (Supplementary Fig. 3c, d) or of mitotic release (Fig. 3g, h). For reconstitution of DENR mutant versions, DENR$^{KO}$ cells were plated at 2 million cells per 10 cm plate, the next day transfected using Effectene (see below). 36 h after transfection cells were re-plated and puromycin (Sigma, #P9620) was added at a concentration of 0.5 µg/ml. 48 h later cells were re-seeded for luciferase assay or Western Blot in puromycin and thymidine-containing medium. 24 h later, during synchronization, medium was exchanged to only thymidine (and no pur-omycin)-containing medium. Synchronization was then completed as described above. Cells were collected for analysis 24 h after end of puromycin selection.

**Transient transfections**. Transient transfections were performed using either Effectene Transfection Reagent (QIAGEN, Cat No./ID: 301425), Lipofectamine 2000 (Thermo Fisher Scientific, Cat No. 11668030) or Lipofectamine 3000 (Thermo Fisher Scientific, Cat No. L300015). The manufacturer's instructions were modified to reduce toxicity and allow ongoing cell cycling and division: For Lipofectamine 2000 transfection (Fig. 4b–g, Supplementary Figs. 9a, b, 4a–e) 0.05 µl of transfection reagent and a total of 20 ng DNA per 96-well were applied 24 h before analysis. For Lipofectamine 3000 transfection (Supplementary Fig. 4f) 0.1 µl of transfection reagent (without enhancer) and a total of 50 ng DNA per 96-well were applied 24 h before analysis. For Effectene transfection (Fig. 3, Fig. 4j, k, Fig. 5c, d, Supplementary Fig. 1b, Supplementary Fig. 3, Supplementary Fig. 6d–f) all transfection reagents as well as the amount of DNA were divided in half and cells were transfected 3 days (Fig. 4, Fig. 5, Supplementary Fig. 6f) or 24 h (all others) before analysis.

**siRNAs and DENR$^{KO}$**. siRNAs were obtained from Dharmacon® (Horizon Dis-covery Ltd.). Catalog numbers and sequences are specified in Supplementary Data 9. For DENR knockdown, a pool of three different siRNAs (−02, −19, −20) was used. siRNA transfection was performed using Lipofectamine RNAi MAX© Transfection Reagent (Thermo Fisher Scientific). HeLa DENR$^{KO}$ cells were gen-erated in our laboratory using CRISPR/Cas9-mediated knockout[32]. Clone 3.42 was used throughout this paper, but effects were observed similarly with clone 2.11.

**Translation/luciferase reporter assay**. To produce reporter constructs, 5′UTR sequences of the indicated genes were PCR-amplified from cDNA or genomic DNA and cloned at the 5′ end of Renilla luciferase using HindIII and Bsp119I.

For the luciferase reporter assays, MCF7 or HeLa$^{WT}$ cells were seeded at a density of 8.000 cells and HeLa$^{DENR\_KO}$ cells at a density of 12.000 cells per 96-well (Supplementary Figs. 4a, 8b, c, 9a, b). For luciferase assays with synchronized U2OS cells (Fig. 4b–g, Supplementary Fig. 4b–f) cells were seeded at a density of 15.000 cells in thymidine. For knockdown experiments, U2OS or HeLa cells were reverse-transfected in 15 cm dishes containing siRNA and Lipofectamine RNAiMAX transfection reagent, and replated 48–72 h thereafter at a density of 10.000 cells per 96-well (all conditions), in thymidine-containing medium if applicable. 16–20 h after (re)plating, cells were transfected via Lipofectamine 2000 with the negative control lamin B-renilla luciferase reporter, or the respective test 5′ UTR-renilla luciferase reporter, as well as a lamin B 5′UTR-firefly luciferase plasmid for internal normalization control. 8–9 h later and 24–28 h after addition of thymidine, cells were released into nocodazole or normal medium (as applicable) as described above. Luciferase activity was analyzed 13 h thereafter, about 24 h after transfection, using DualGlo Luciferase Assay System (Promega) and Mithras LB 940 Reader (Berthold Technologies). For asynchronous cells all medium changes and washes were performed simultaneously, without the addition of synchronizing agents.

For analysis of results, in a first step, renilla luminescence was normalized to firefly luminescence to control for transfection/expression variabilities. In a second step, this ratio for the 5′UTR reporter of interest was normalized to the equivalent ratio of the negative control lamin B-RLuc/lamin B-FLuc reporter from parallelly-transfected and equally-treated wells ("normalized RLuc/FLuc"), to control for variability between conditions.

**Cell lysis and immunoprecipitation**. For cell lysis, cells were scraped, spun down at 13.000 × g and resuspended in lysis buffer containing Tris /HCl pH 7.5 50 mM, NaCl 250 mM, EDTA 1 mM, Triton X-100 0,1%, NaF 50 mM, protease inhibitors (cOm-plete Protease Inhibitor Cocktail, Sigma Cat No. 4693116001, 1 tablet in 10 ml) and phosphatase inhibitors (sodium orthovanadate 1 mM, β-glycerophosphate 100 mM, PhosSTOP EasyPAK (Sigma, Cat No. 4906845001) 2 tablets in 10 ml). After 10 min incubation on ice, samples were centrifuged for 10 min at 20,800 g at 4 °C. For immunoprecipitation, supernatant was divided into lysates and samples for immu-noprecipitation. The latter were incubated with anti-FLAG M2 affinity gel (Sigma, Cat No. A2220-1ML) for 1.5 h at 4 °C (FLAG-IP). For washing the immunoprecipitated proteins, beads were centrifuged for 1 min at 200 × g, supernatant was removed, and fresh lysis buffer was added to the beads. This step was repeated three times before the beads were transferred to a new epi and elution was performed with 1× Laemmli buffer. For the whole-cell lysates, their protein concentration was measured using the biorad protein assay (biorad, Cat No. 500–0006) and adjusted before addition of 5xLaemmli buffer.

**Western blots**. Equal protein amounts were run on SDS-PAGE gels and trans-ferred to nitrocellulose membrane with 0.2 µm pore size. After Ponceau staining, membranes were incubated in 5% skim milk PBST for 20–60 min, briefly rinsed with PBST, and then incubated in primary antibody solution (5% BSA PBST or 5% skim milk PBST) overnight at 4 °C. Membranes were then washed three times, 15 min each in PBST, incubated in secondary antibody solution (1:10,000 in 5% skim milk PBST) for 1 h at room temperature, then washed again three times for 5–15 min. Finally, chemiluminescence was detected using ECL reagents and detected using a Biorad ChemiDoc imager. For immunoblot visualization and analysis ImageLab software version 5.2.1 was used.

**Antibodies**. The following antibodies were used: ABLIM1 (WB: 1:1000, IF: 1:500, rabbit, bethyl-biomol A302-237-T), β-Actin (1:5000, mouse, Sigma #A2228), ATF-4 (1:1000, rabbit, Cell Signaling #11815), γ-tubulin (1:1000, mouse, Abcam #ab27074), Caspase 3 (1:1000, rabbit, Cell Signaling #9662), CDKL5 (WB: 1:1000, IF: 1:500, rabbit, abcam ab22453), CDT1 (1:1000, rabbit, Cell Signaling #8064), cleaved Caspase 3 (WB: 1:1000, flow cytometry 1:100, rabbit, Cell Signaling #9664), CUL1 (1:500, mouse, Invitrogen #32-2400), cyclin A2 (1:1000, rabbit, Cell Sig-naling #91500), cyclin B1 (1:1000, mouse, Cell Signaling #12231), cyclin E1 (1:1000, rabbit, Cell Signaling #20808), DENR (WB and IF: 1:2000, guinea pig, in-house production), pDENR_Ser73 (WB: 1:500, IF: 1:200, rabbit, custom-made by inno-vagen AB, Lund, Sweden), DUSP4 (WB: 1:1000, IF: 1:500, rabbit, abcam ab216576), FLAG (1:1000, rabbit, Sigma #F7425), FLAG-M2 (1:1000, mouse, Sigma #F3165), GAPDH (1:2000, rabbit, Cell Signallling #2118), Geminin (1:1000, rabbit, Cell Signaling #52508), pHH3 (Serine 10) (for WB: 1:500, rabbit, Cell Signaling #9701; for flow cytometry: 1:100, mouse, Cell Signaling #9706 S), MAP2K6 (WB: 1:1000, IF: 1:500, rabbit, Cell Signaling #8550), MCTS1 (1:1000, guinea pig, in-house production) a-raf (1:1000, rabbit, Cell Signaling #4432), c-raf (1:1000, rabbit, Cell Signaling #9422), ERK1/2 (1:1000, rabbit, Cell Signaling #4695), GST-HRP conjugate (1:5000, goat, GE Healthcare #RPN1236), pERK1/2 (T202/Y204) (1:1000, rabbit, Cell Signaling #4370), p-p90RSK (T359/Y363) (1:1000, rabbit, Cell Signaling #9344),RSK1/2/3 (1:1000, rabbit, Cell Signaling #14813).

## In vitro kinase assay

*Non-radioactive.* 50 ng of purified His-tagged DENR·MCTS1 protein complex was incubated with 25 ng of active CDK1-Cyclin B1, CDK2-Cyclin A2, CDK2-Cyclin E1, CDK4/Cyclin D1, CDK6/Cyclin D1 proteins (Proqinase #0134-0135-1, #0050-0054-1, #0050-0055-1, #0142-0143-1, #0051-0154-2) in Protein Kinase-Buffer (50 mM HEPES-NaOH pH 7.5, 3 mM $MgCl_2$, 3 µM Na-orthovanadate, 1 mM DTT) with 1.25 µM ATP for 30 min at 30 °C. Then Laemmli buffer was added and half of each sample analyzed by Western blotting using phospho-specific antibodies.

*Radioactive.* Protein kinase assays were carried out in 20 µl kinase buffer (60 mM HEPES-NaOH (pH 7.5), 3 mM $MgCl_2$, 3 mM $MnCl_2$, 3 µM Na-orthovanadate, 1.2 mM DTT, 50 µg/ml PEG20.000, 5 µCi of [γ-32P] ATP (Perkin Elmer (NEG502A)) and 40 µM [γ-S]-ATP either with the enzyme only, or 1 µg of the respective substrate. As positive control, 1 µg Casein was used. First, samples were incubated for 30 min at 30 °C. The reaction was stopped by the addition of 5 µl 5× SDS-sample buffer and incubation at 95 °C for 5 min. The proteins were separated by SDS-gel electrophoresis (12% PAA). Subsequently, SDS-gels were stained with Coomassie Staining solution (0.4% Coomassie Brilliant Blue G250, 10% citric acid, 8% ammonium sulfate, 20% methanol) for 2 h and destained with several washes of 20% Methanol in $ddH_2O$ until the background was clear. Finally, the SDS-gels were placed on thick filter paper and dried with a vacuum Gel-Drying System for 1.5 h at 80 °C. Radioactive signals were visualized by exposure to X-ray films.

**Immunofluorescence.** For immunofluorescence microscopy, cells were plated on poly-L-lysine (Sigma # P4707)-coated cover slips and one to 3 days later fixated with PFA 4% for 20 min at room temperature. After washing four times with PBT (Triton X-100 0.1% in PBS), samples were blocked with BBT (PBT with BSA 1%) for 2 h and incubated with primary antibody overnight. Anti-pDENR (Ser73) purified antibody was diluted in BBT 1:200, anti-DENR antibody was diluted in BBT 1:1000, Anti-γ-tubulin antibody (Supplementary Fig. 7c) was diluted in BBT 1:1000. Next, samples were washed four times with BBT and incubated with secondary antibodies (AlexaFluor488 rabbit and Alexa Flour594 goat (Life Technologies)) at a dilution of 1:1000 in BBT for 2 h. Then samples were washed four times with BBT and during the third wash DAPI was added at a dilution of 1:1000. After the fourth wash, cells were equilibrated in mounting medium for 10 min or overnight and then mounted. Images were taken using a Leica SP8 Confocal Laser Scanning Microscope.

**Flow cytometry.** Proliferating HeLa^DENR_WT or HeLa^DENR_KO cells were trypsinized, collected, washed once with cold PBS, spun down for 5 min at 400 × g, taken up in 600 µl of cold PBS and slowly dropped into an epi containing 1.2 ml of ice-cold ethanol 70% while maintaining a constant vortex. Cells were then fixed at −20 °C overnight. Next, cells were spun down at 600 × g for 10 min and washed twice with PBS. The cell pellet was carefully resuspended in PBS + Triton X-100 0.2% and gently rocked on ice for 15 min to allow permeabilization. Cells were washed once more in PBS, taken up in 200 µl BSA 1% containing anti-pHH3 and anti-CC3 primary antibodies and gently rocked at 4 °C overnight. Next, the pellet was washed once more with PBS and then incubated with fluorescent secondary antibodies (see above, immunofluorescence) in BSA 1% gently rocking for 30 min in the dark. After a final PBS wash the pellet was taken up in PBS and analyzed using a Guava® easyCyte™ flow cytometer running Guava Soft 3.3.

**Ribosome profiling.** HeLa wildtype and DENR^KO (clone 3.42 or clone 2.11) cells were seeded in 15 cm dishes at 1 million cells per dish (DENR^WT) or 1.5 million cells per dish (DENR^KO) in 20 ml growth medium. Mitotically enriched cells were synchronized as described above and asynchronous cells were left untreated and harvested 2 days later for Ribo-seq and RNA-seq. For asynchronous cells two 15 cm dishes per condition were used for RiboSeq and one 10 cm dish for RNASeq. For mitotically enriched cells, four 15 cm dishes per condition were used for RiboSeq and one 15 cm dish per condition was used for RNASeq. In the asynchronous sample, mitotic shake-off was performed before harvest to de-enrich for mitotic cells and supernatant was discarded. Then cells were washed and lysed as described below. In the mitotically enriched sample, mitotic cells were shaken off and spun down at 300 × g, and remaining cells were washed and both cell pools (shaken-off and on-plate) were lysed together as described below. For RNA-seq, cells were lysed with TRIzol and total RNA was isolated following manufacturer's protocol. For Ribo-seq, cells were briefly rinsed with ice-cold PBS containing 10 mM MgCl2, 400 µM cycloheximide (CHX), then spun down again at 300 × g and lysed with 150 µl of lysis buffer (250 mM HEPES pH 7.5, 50 mM MgCl2, 1 M KCl, 5% NP40, 1000 µM CHX) per sample. After brief vortexing, lysate was clarified by centrifuging for 10 min at 20.000 x g at 4 °C. Approximate RNA concentration was measured using a Nanodrop and 1 µl of Ambion RNAse 1 was added per 120 µg of measured RNA. Lysates were incubated with RNAse for 5 min on ice. Lysates were then pipetted onto 17.5–50% sucrose gradients and centrifuged at 155.000 × g for 3.5 h in Beckmann SW40 rotor. Gradients were fractionated using a Biocomp Gradient Profiler system and 80S fractions were collected for footprint isolation. RNA was isolated from these fractions using Acid-Phenol extraction and analyzed on an Agilent Bioanalyzer system to asses RNA integrity.

**Deep-sequencing library preparation.** RNA samples were depleted of ribosomal RNA using the Illumina Ribo-Zero Gold kit. Depleted total RNA was then fragmented using chemical cleavage in 50 mM NaHCO3 at pH 10, 95 °C for 12 min. Then total RNA was processed in parallel with the depleted RNA from 80S ribosome fractions. For size selection, RNA was run on 15% Urea-Polyacrylamide gels and fragments from 25–35 nt were excised using reference ssRNA nucleotides of 25 and 35 basepairs run on a neighboring lane. RNA was extracted from the gel pieces and phosphorylated using T4 PNK. Deep sequencing libraries were prepared from these RNA fragments using the Bio-Scientific NEXTflex Small RNA-Seq Kit v3. Deep-sequencing libraries were sequenced on the Illumina Next-Seq 550 system.

**Data analysis.** Adapter sequences and randomized nucleotides were trimmed from raw reads using cutadapt. rRNA and tRNA reads were removed by alignment to human tRNA and rRNA sequences using bowtie2 version 2.3.4.2. Then, the remaining reads were aligned to the human transcriptome using BBmap. Metagene plots, single transcript traces and grouped analyses were carried out or created with custom software written in C available at https://github.com/aurelioteleman/Teleman-Lab and the Zenodo repository (DOI 10.5281/zenodo.5751288).

**Statistics.** Theoretical Z vs Observed Z analysis was carried out in excel. The statistical tests used are indicated in the figure panels.

**Software.** DNA sequence analysis was done with A Plasmid Editor (ApE) v2.0.49.10. Statistical analyses were done with GraphPad Prism version 9.0.0 (86). Analysis of flow cytometry data were done with Guava Soft 3.3.

**Reporting summary.** Further information on research design is available in the Nature Research Reporting Summary linked to this article.

## Data availability

The riboseq and RNA-seq datasets in this study have been deposited at NCBI SRA under accession code PRJNA768478. The riboseq and RNASeq source data generated in this study are provided in Supplementary Information. Source data are provided with this paper.

## Code availability

All custom software used in this study is available at GitHub: https://github.com/aurelioteleman/Teleman-Lab and in the Zenodo repository (https://doi.org/10.5281/zenodo.5751288)[65].

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

## Acknowledgements

This work was supported by a DFG grant (project number 316695455) to A.A.T., by the DFG SFB 1036 to A.A.T., by a DKFZ NCT3.0 Integrative Project in Cancer Research (NCT3.0_2015.54 DysregPT) grant to A.A.T., and by a Cell Networks—Cluster of Excellence (EXC81) grant to K.C.v.H.

## Author contributions

K.C.v.H., S.M., S.S., and M.M. performed experiments with help from J.B., K.C.v.H., M.M., T.G.H., and A.A.T. designed the work. K.C.v.H., S.M., S.S., M.M., J.B., T.G.H., and A.A.T. analyzed and interpreted the data. K.C.v.H. and A.A.T. wrote the manuscript.

## Funding

## Competing interests

The authors declare no competing interests.
