## [Peer Review File · Nature Communications]

Cyclin B/CDK1 and Cyclin A/CDK2 phosphorylate DENR to promote mitotic protein translation and faithful cell divisionREVIEWER COMMENTS

Reviewer #1 (Remarks to the Author):

The manuscript „Cyclin A/CDK2 phosphorylates DENR to orchestrate mitotic protein translation and faithful cell division „ by von Hohenberg and colleagues propose that CyclinA/CDK2-dependent phosphorylation of Serine 73 at an oncogenic protein DENR regulates DENR•MCTS1 (a protein complex that is involved in non-canonical translation initiation) by promoting the stability of DENR protein and enhanced translation of mRNAs required for mitosis.

The study uses phospho-specific antibodies to detect S73 phosphorylation using immuno-precipitations, western blotting, and immuno-stainings to confirm the phosphorylation status and dynamics during the cell cycle in U2OS cells. For dissecting kinase specificity they use CDK inhibitors and for proteolytic function the caspase inhibitor.

In general, the finding is potentially interesting and important in the context of cell cycle regulation and future cancer drug targeting strategies. However, the manuscript can be considerably improved, and at the current state, I cannot recommend it for publishing due to several questions and methodological shortcomings outlined below.

Major points

1) The authors claim that Ser73 is positioned within a CDK2/Cyclin A target consensus motif (Suppl. Figure 3a) and they cite Stevenson-Lindert et al. 2003 JBC (reference 44). The sequence presented in Suppl. Figure 3a, the (68) LTVENSPKQEA (80) does not match Cyclin A consensus sequence according to the cited paper (also the numbering in Suppl. Figure 3a is wrong, instead of 80, it should be 78). Instead, Stevenson-Lindert et al. demonstrate that sequence PKTPKAAKKL, where similarly to Ser73, the basic specificity determinant +3K/R is not present, exhibits K_M of 2500 micromolar, while the parent peptide PKTPKAKKL shows K_M of 20 micromolar. Thus, the +2K is not a particularly important determinant alone for the specificity of Cyclin A/CDK2 complex, and one cannot say that LTVENSPKQEA matches the consensus sequence of Cyclin A, particularly. Also, the authors do not consider or discuss that Ser73 could be a good or even better phosphorylation motif for CyclinB/CDK1 (see also below).

2) Suppl. Figures 3b and 2 are not sufficient to make a claim „data indicate that DENR Ser73 is a direct substrate of CDK2/Cyclin A2 in early mitosis“. The authors should provide more evidence that cyclin B is not involved in this phosphorylation. In a couple of places in the manuscript, the authors mention that they show a nice example of cyclin specificity, but this was only studied relatively to cyclin E. The general active site-specificity of CDK complexes is increasing gradually in the cell cycle and it is not clear why cyclin B was excluded from the model without testing. Such gradual activity pattern was shown in mammalian cells by Skotheim and colleagues (Topacio et al. Mol Cell (2019) 74) and was first discovered in vitro by Loog and colleagues in Kõivomägi et al. Mol Cell (2011) 42, in *S.cerevisiae*, and later

confirmed in vivo in Örd et al. Nat Struct Mol Biol (2019) 26. These works should be cited, and a similar in vitro kinase assay panel with different cyclins, as presented in these works, should be presented to make clear, which kinase complex is responsible for the phospho-regulation mechanism described in the paper. Otherwise, the claim that Ser73 is a target of CDK2/Cyclin A2 alone and only, like stated even in the title (!), would be entirely misleading and wrong.

3) The authors should provide clear references or experimental data (IC50 or Ki values) that seliciclib or A-674563 are exclusively CDK2 inhibitors and that inhibition of CDK1 is not the case in vivo, even partially. Alternatively, again, we have no proper answer to what kinase complex are we talking about. Even a minor effect on CDK1 can seriously change the interpretation of the results and the model proposed in the study.

4) The effect in Figure 3e is based on very faint bands, but the driven conclusions are strong. The authors should quantify the bands from repeated replica experiments to make a stronger case.

5) Figure 3f,g seems to contradict Figure 1a-c concerning the ratio of phosphorylated and non-phosphorylated form.

6) The authors could discuss the effect of MCTS1[T117A,S118A] mutations in relation with DENR S73A mutation. Is the double site constantly phosphorylated or oscillates in a cell cycle-dependent manner as well?

7) The authors raised antibodies against „six predicted phospho-sites in DENR and MCTS1“. However, it is not explained how these sites were predicted. Similarly, on Page 7:“ Hence we mutated to alanine all serines and threonines in DENR and MCTS1 that were observed or predicted to be phosphorylated....“. Please define better how they were observed or predicted and provide a list (some short table that would be easier to grasp than the Excel files in the Supplementary section).

8) The authors mention that only the DENR Set73 phospho-antibody was generated successfully. But what about the others? Was this the reason why other candidate sites were excluded from the further study?

9) The authors mention briefly that mass spectrometry was performed on DENR-MCTS1 complex immunopurified from untreated HeLa cells and observed phosphorylation of S73 and T69. The latter site was marked „not shown“. It is not clear why T69 was not studied further, or at least explain why they were not interested in it. Was it quantitative mass spectrometry and were these two sites the only sites that were phosphorylated at the complex? What about the previously mentioned MCTS1-T117,S118?

10) PLK kinases may use CDK phosphorylation sites as primers. The authors could discuss this possibility, especially that they use a substrate complex produced using the bacterial system for in vitro kinase assay (Suppl Figure 2c) that would lack any priming phosphorylations.

Minor points

1) Page 7, please explain in the main text what the „activity ratio >3-fold autophosphorylation“ exactly means: „...identified 50 kinases with the capacity to phosphorylate DENR•MCTS1 in vitro at an activity ratio >3-fold autophosphorylation...“

2) Page 7, wording too bold: just knocking down kinase expression does not provide such a validation ...“We further validated these hits by testing whether any of these kinases regulate DENR•MCTS1 activity in vivo by knocking down their expression and assaying stuORF reporter activity in two cell lines.

3) Page 44 Suppl Figure 2c legend: „DENR•MCTS1 complex or one of its mutants were purified from bacteria“. This sounds like they are bacterial proteins. „Bacterial expression system“ would be more precise.

4) The sentence: „We exposed cells to different stresses and environmental conditions, including tunicamycin to induce ER stress, high vs. low cell density, staurosporin to induce apoptosis, doxorubicin to induce DNA damage and amino acid and glucose starvation, and in this way discovered that DENR is phosphorylated at Ser73 in a cell cycle-dependent manner (Figure 1):... is unclear, as the results of the experiments performed in all these different conditions are not shown in Figure 1, and it seems like these listed conditions somehow led to the knowledge that Ser73 was phosphorylated in cell cycle-dependent manner and that the following double thymidine block-release experiment was done just to further confirm it.

Reviewer #2 (Remarks to the Author):

In their manuscript entitled “Cyclin A/CDK2 phosphorylates DENR to orchestrate mitotic protein translation and faithful cell division”, Clemm von Hohenberg et al. expand on findings reported in several previous studies from the same lab, which have established the non-canonical translation initiation factor DENR-MCTS1 as a regulator of mRNA translation for specific subsets of transcripts that contain regulatory upstream open reading frames (uORFs). On these transcripts, DENR-MCTS1 mediates re-initiation after uORF translation. Previous work by the Teleman lab and others was dedicated mainly to the identification of (i) DENR-MCTS1 client transcripts, (ii) of the functions and pathways that these transcripts are associated with, and (iii) of the characteristics of the responsible uORFs. In particular, these previous studies established a link between DENR-MCTS1/uORF re-initiation and tissue growth, as well as cancer. Several studies have established a pro-cancer, oncogenic role of DENR-MCTS1.

The current manuscript now addresses a novel aspect of DENR-mediated gene expression control, which is the post-translational regulation of DENR itself. The authors use their findings to present a model according to which DENR is phosphorylated through CDK2/CycA at Serine 73 during the cell cycle. This phosphorylation is observable during M-phase (deposited at mitotic onset; removed at exit), where it is necessary to protect DENR from caspase-mediated proteolytic degradation. Indeed, reporter assays based on several previously identified, cell cycle-relevant DENR targets show preferential DENR regulation in mitotic cells as compared to cells grown under asynchronous conditions. Finally, in DENR knock-out cells (or in a DENR S73A mutant), there is high prevalence of anaphase catastrophe and cell death, presumably due to the lack of translation of the relevant mRNAs.

The presented data are intriguing and novel, of generally good quality, and on a high-interest topic; the described links with the cell cycle could further rationalise some of the previously reported links between DENR and cancer. I therefore encourage the authors to improve the manuscript - as described in two main points and additional other points, below - to bring it in shape for a publication.

In my view, the main weakness of the paper is that it currently does not offer an answer or plausible explanation to the question why this intricate regulation of DENR in the cycle is needed, i.e. why can DENR apparently be degraded cell cycle-dependently, just to then get “rescued” from the degradation through CDK2-dependent phosphorylation, making DENR abundance finally more-or-less constant over the day? Is this rescue occurring constitutively? Or is the cell cycle-dependent degradation actually sometimes occurring on the WT protein, and under which conditions and with what consequences for the translome? Another conceptually important, open question (related to the previous one) is whether the functional importance of phosphorylation lies in stabilising DENR, or whether it also reprograms the specificity of re-initiation to specific substrates in mitosis as compared to interphase? I believe that with the data in Fig. 4, the authors want to make this claim, but in reality one would need comprehensive evaluation of translation rates (ribosome profiling) to be sure. I think this is the one and only, main experiment that would solve many of my questions, and I am therefore suggesting it below as Main point 1.

Main points:

(1) Does the phosphorylation only act through the regulation of DENR abundance, or is there also an aspect of reprogramming of specificity involved? In Figure 4, many questions remain open. Are the reporters mitotically DENR-reinitiated because they are “special” in something (uORFs?), or because there is something cell cycle-specific about the expression of their RNAs (e.g. RNA abundance regulated across cell cycle, possibly even through uORF and NMD that would also be active on reporter mRNAs) ? The reporter assays as they are currently shown do not give much information on these possibilities; e.g. there are no control reporters with mutated uORFs; there are also no statistics (p-values) shown in Fig. 4B-C.

Suggestion: Why not do ribo-seq on asynchronous and mitotic cells, expressing WT DENR vs. KO (or vs. DENR S73A) ? I would even recommend also including the S73D mutant (phospho-mimic). I am convinced that such an experiment would solve many of the open questions of the study. It would establish whether endogenous DENR client mRNAs are regulated depending on phospho-status and depending on mitotic phase.

(2) Another experiment that should distinguish between functional reprogramming vs. simple abundance effects, could be a mutant in which S73A is combined with a mutation in the caspase site (i.e., mentioned in the manuscript to be at around amino acid 26 - according to the Steitz lab structure, it should be possible to point-mutate this area without affecting interaction with MCTS1 or the ribosome). This should rescue the low stability of S73A alone. Can the authors make and test such a mutant in some of their assays - e.g. stability (like Fig. 3C-D) and functional effects (e.g. assays as in Fig. 5)?

Other Points:

(3) Figure 2: In panel A, the CDK2-Cyclin A2 vs. CDK2/Cyclin E1 effect is beautiful and shows extraordinary specificity - rare to see this so clearly in an in vitro assay! I am a little more concerned about the in vivo data and the specificity of the drugs at the concentrations used. For example, A-674563 inhibits many kinases in the nM range (see e.g. <https://www.selleckchem.com/products/a-674563.html>: Akt1, PKA, GSK3b, ERK2 all inhibited). For Seliciclib, it seems to be a bit better, but still - the 10 uM concentration is high (close to the IC50 of, for example, ERK2, 14 uM). At lower concentrations, well-above the IC50 for CDK2, the effect on pDENR is not impressive (1 uM Seliciclib; 0.2 uM A-674563; Figure 2B). Can the authors comment on the specificity of the drugs?

(4) Do the authors have any information on the approximate fraction of DENR that is phosphorylated in mitosis?

(5) Figure 3: In panel A and B: The input image for DENR shows in EV cells high levels of endogenous DENR. Hence, I assume these experiments were not done in DENR knockout cells (it's not described in the legend) - however, wouldn't that have been a better genetic background for these experiments?

(6) Figure 3C/D: Western blots are notoriously difficult to quantify, as the signal does not scale linearly with protein amount. Comparing/evaluating the wt and mutant time course as it is shown in the western blot is difficult.

Could the authors show an additional, longer Western exposure for the mutant, in which the "0" time-point is similarly strong as the 0 time-point of the WT? That would make it easier to evaluate if the quantification in panel D corresponds to what one sees in C.

(7) Figure 3F: I find the DENR rescue by Z-VAD-FMK not very convincing. For a start, more replicates and quantification would be needed. Maybe the best experiment would be the mutation of the caspase site, see above point (2).

(8) Figure 5D: statistics should be added. Do the differences pass the typical tests for statistical significance?

(9) Finally: The structure of the manuscript is a little bit awkward and could be much improved, especially at the beginning. For example, for the entry to the results section, there are a lot of "negative results" (the various phospho-site mutants) that are not of immediate further relevance to the overall narrative – they take a lot of space and distract from the main thrust. Some of this is confusing for the beginning of a paper.

Example on page 5: "Worth mentioning is that we performed these experiments in HeLa cells in standard culture conditions, where many of these residues may not be phosphorylated. Hence, in the future it may be worth testing this panel of mutants also in other cell lines or environmental conditions. Furthermore, we used asynchronous cells, explaining in part why the DENR S73A mutant, which is relevant only in mitosis (see below), did not have a strong phenotype." – some of these comments only become clear later in the manuscript. I believe this actually destabilises, rather than helps, the reader here.

Moreover, just after the part cited above, the description how the authors finally settled on studying Ser73 reads very arbitrary - and although this may honestly be how the project in reality developed, I recommend presenting in a slightly more targeted fashion.

Thus, I would suggest restructuring and/or describing some of the less relevant experiments (read: less relevant for the actual story the authors want to tell) much shorter in the main text, and moving a lot of the description/interpretation to the supplementary figure legend or, in part, to discussion. I still think the "negative" data should be reported (it may indeed be helpful for others), but it should take less space and be not so prominently placed.

In general, the introduction and discussion could also profit from a bit better integration and flow in the arguments.

In summary: The data could be much more effectively presented.

Reviewed by David Gatfield (assisted in reviewing by PhD student Romane Meurs). In case any of the above points are unclear, I encourage the authors to contact me directly.

Reviewer #3 (Remarks to the Author):

Hohenberg et al address whether and how activity of the DENR-MCTS1 complex is regulated by posttranslational modifications. They report a comprehensive analysis of modifications and assess their functional significance in appropriate mutants. They find that DENR undergoes CDK2-dependent phosphorylation on Ser73 in mitosis and suggest that this modification is important to promote translation of specific mitotic target genes and error-free mitoses.

This is an interesting model and a useful contribution to the emerging view that selective translational regulation of proteins involved in mitosis is essential for faithful, correct mitosis. Although the phenomenon of cell-cycle-specific translation has been described by a number of labs, the mechanisms behind it are not well understood.

They have provided compelling evidence that DENR-S73 phosphorylation is cell-cycle dependent and high in mitosis; that DENR is phosphorylated on S73 by CDK2-CycA in vitro and in vivo; and this phosphorylation does not affect the interaction with MCTS1 but affects the stability of DENR in mitosis.

They then explore whether DENR-S73 phosphorylation has an effect on mitotic translation and thereby on mitosis. They use reporter constructs using the 5'UTR-s of selected DENR target genes and show that DENR is required for their optimal translation. They also assess the levels of some of the endogenous

proteins with known mitotic functions. Finally, they show that in the absence of DENR cells accumulate in mitosis and have aberrant mitoses.

Overall, the manuscript is very clearly written and the data are consistent with the model. However, I believe some control experiments and data analyses would strengthen the conclusions.

1) Fig 1 d : The localization of P-DENR in mitotic cells was investigated. The authors claim that the P-DENR antibody lights up the contractile ring in telophase. A good control would be the KO transformed with DENR-S73A. Notably, the midbody is notorious for staining positive with a number of phospho-specific antibodies. Furthermore, according to their model DENR is degraded in a caspase-dependent manner in anaphase -telophase, making it less likely that the brightly stained signal in telophase is due to DENR. On the other hand, if it is, it supports their suggestion that there is local translation at this site or that DENR may have a function other than translation.

2) The putative target proteins were not enriched in mitosis as much in KO as in wt, this phenotype was rescued by transfection with wt DENR – but rescue should be addressed also with the S73A mutant, either using the reporter and/or measuring endogenous protein levels. Specifically, at least some of the experiments in fig 4 and 5 should be repeated in a KO transfected with the S73A mutant. The only phenotypic analysis with the DENR-S73A mutant is shown in fig 5d, but as discussed in point 4-5, the analysis of mitotic progression could be improved.

3) U2OS cells were used in figures 1-4, then in Fig 5 HeLa cells were used to show an accumulation of cells in mitosis in DENR KO. Is this phenomenon specific for HeLa cells? Why is the model system changed? In general, it would be reassuring to see some of the key experiments performed in more than one cell line, and perhaps also in a non-transformed cell line.

4) Fig 5d shows two biological repeats and assessing the stages of mitosis based on DAPI staining. This is very difficult to do based on DAPI alone and if one is to conclude anything about the problems in mitosis, better images and a better approach are required. See also point 5.

5) In the Discussion the authors describe the phenotype upon DENR KO as anaphase catastrophe, largely because of the involvement of CDK2 in DENR phosphorylation. However, anaphase catastrophe is characterized by multipolar spindles rather than “mitotic blebs” and in this work it is not addressed in detail how the cells die, only that they die in mitosis. Therefore it is not correct to state that “DENR loss of function phenocopies this mitotic failure”(ie anaphase catastrophe). Even though the model of anaphase catastrophe is an attractive one, they should demonstrate it by eg live-cell imaging or immunofluorescence and quantification of multipolar spindles. At the very minimum, they should reformulate the statement and describe the phenotype as aberrant mitoses or similar.

6) An overlap between the DENR-dependent transcripts and transcripts whose translation is increased in mitosis would strengthen the model. The datasets available from the literature – and cited in the manuscript (5, 8) – and their datasets on DENR-dependent transcripts should be compared. See also point 7.

7) The finding that the synthetic stuORF-reporter does not show DENR-dependent increased translation in mitosis is rather puzzling, and it would be interesting to have the authors' view on the possible reason. They even report an increase of the translation in mitosis of DENR-dependent genes that have not been associated with mitosis (Fig 4), but not of the synthetic reporter. Does the synthetic stuORF lack something that the 5'UTR-s of the mitotic genes have? Do they have anything in common other than the presence of small uORF-s? Or is it that even though DENR phosphorylation and stabilization correlate with mitotic translation, it is not sufficient for the mitotic DENR-targets being efficiently translated? This issue brings us back to the importance of performing the rescue experiments with the DENR-S73A mutant suggested in point 2.

Minor comments:

1) In the Methods section the nocodazole concentration is described as 400 ng/ml. This seems very high, even 40 ng/ml is sufficient to arrest the cells at G2/M. Although this is unlikely to change the conclusions, such a high concentration would be a severe stress to the cells, leading to some cells arresting in interphase, and this should be considered when interpreting the results. Unless it was just a typo.

2) "U2OS or HeLa cells were plated in 15cm dishes on siRNA and Lipofectamine RNAiMAX transfection reagent" - something must be missing here.

Reviewer #1

The manuscript „Cyclin A/CDK2 phosphorylates DENR to orchestrate mitotic protein translation and faithful cell division „ by von Hohenberg and colleagues propose that CyclinA/CDK2-dependent phosphorylation of Serine 73 at an oncogenic protein DENR regulates DENR•MCTS1 (a protein complex that is involved in non-canonical translation initiation) by promoting the stability of DENR protein and enhanced translation of mRNAs required for mitosis. The study uses phospho-specific antibodies to detect S73 phosphorylation using immuno-precipitations, western blotting, and immuno-stainings to confirm the phosphorylation status and dynamics during the cell cycle in U2OS cells. For dissecting kinase specificity they use CDK inhibitors and for proteolytic function the caspase inhibitor.

In general, the finding is potentially interesting and important in the context of cell cycle regulation and future cancer drug targeting strategies. However, the manuscript can be considerably improved, and at the current state, I cannot recommend it for publishing due to several questions and methodological shortcomings outlined below.

We thank the reviewer for the encouraging evaluation and the insightful comments. We have included a significant amount of new data to address the reviewer's comments, which we think they have made the manuscript significantly stronger.

Major points

1) The authors claim that Ser73 is positioned within a CDK2/Cyclin A target consensus motif (Suppl. Figure 3a) and they cite Stevenson-Lindert et al. 2003 JBC (reference 44). The sequence presented in Suppl. Figure 3a, the (68) LTVENSPKQEA (80) does not match Cyclin A consensus sequence according to the cited paper (also the numbering in Suppl. Figure 3a is wrong, instead of 80, it should be 78). Instead, Stevenson-Lindert et al. demonstrate that sequence PKTPKAAKKL, where similarly to Ser73, the basic specificity determinant +3K/R is not present, exhibits K_M of 2500 micromolar, while the parent peptide PKTPKKAKKL shows K_M of 20 micromolar. Thus, the +2K is not a particularly important determinant alone for the specificity of Cyclin A/CDK2 complex, and one cannot say that LTVENSPKQEA matches the consensus sequence of Cyclin A, particularly. Also, the authors do not consider or discuss that Ser73 could be a good or even better phosphorylation motif for CyclinB/CDK1 (see also below).

(We have fixed the numbering in Suppl. Figure 3a (now Suppl. Figure 2a) – thanks for catching that.)

We thank the reviewer for this comment which prompted us to do a significant number of additional experiments, testing CDK1 both *in vitro* and *in vivo*. As the reviewer suspected, we find that CDK1 phosphorylates DENR Ser73 even better than CDK2 *in vitro* (new Figure 2a-b) and that pharmacological inhibition of CDK1 can cause a larger drop in pDENR(Ser73) *in vivo* than CDK2 inhibition (Figures 2c-f and Suppl. Figures 2b-c). The relative contribution of CDK1 and CDK2 to Ser73 phosphorylation *in vivo* seems to depend on cell line, with CDK1 predominating in HeLa cells (Figure 2c-d vs Suppl. Figure 2g) and CDK2/CycA predominating in U2OS cells (Figure 2e-f vs Suppl. Figure 3b).

The *in vivo* experiments were tricky to do because CDK1 activity is required for entry into mitosis. To test the contribution of CDK1 to DENR phosphorylation we first synchronized cells at the G2/M transition with a CDK1 inhibitor, then released this block, allowing cells to enter mitosis (e.g. causing an increase in pDENR, lane 3 in Figure 2c), and then briefly re-applied the CDK1 inhibitor RO3306 (Figure 2c-d). In contrast, the CDK2 inhibitor does not block mitotic entry. Therefore, to test the contribution of CDK2, including in early mitotic phases which could be missed by the approach mentioned above, we synchronized the cells in S phase and then released the cells so they could enter into mitosis, and briefly applied CDK2 inhibitors in late G2 (Figure 2e-f). Alternatively, we used an asynchronous population of cells, treated only briefly with CDK2 inhibitor and looked at pDENR by staining cells so we could visually identify the cells that were in mitosis via DNA morphology (Suppl. Figure 2d-f).

In the original submission, we had excluded CDK1 as an upstream kinase because we had added RO3306 to an asynchronous population of cells and assayed DENR phosphorylation by western blot. This causes a drop in phospho-histone H3 (as a control for drug efficacy) but not in DENR phosphorylation. It is only using this more careful analysis described above that the contribution of CDK1 becomes evident.

Hence, thanks to the reviewer's comment, we now conclude that both CDK1 and CDK2 phosphorylate DENR Ser73, and have changed the Results, Discussion and the title of the manuscript.

2) Suppl. Figures 3b and 2 are not sufficient to make a claim „data indicate that DENR Ser73 is a direct substrate of CDK2/Cyclin A2 in early mitosis“. The authors should provide more evidence that cyclin B is not involved in this phosphorylation. In a couple of places in the manuscript, the authors mention that they show a nice example of cyclin specificity, but this was only studied relatively to cyclin E. The general active site-specificity of CDK complexes is increasing gradually in the cell cycle and it is not clear why cyclin B was excluded from the model without testing. Such

gradual activity pattern was shown in mammalian cells by Skotheim and colleagues (Topacio et al. Mol Cell (2019) 74) and was first discovered *in vitro* by Loog and colleagues in Kõivomägi et al. Mol Cell (2011) 42, in *S.cerevisiae*, and later confirmed *in vivo* in Örd et al. Nat Struct Mol Biol (2019) 26. These works should be cited, and a similar *in vitro* kinase assay panel with different cyclins, as presented in these works, should be presented to make clear, which kinase complex is responsible for the phospho-regulation mechanism described in the paper. Otherwise, the claim that Ser73 is a target of CDK2/Cyclin A2 alone and only, like stated even in the title (!), would be entirely misleading and wrong.

We thank the reviewer for this comment. As detailed in the response to issue #1 above we have now tested CDK1/CycB both *in vitro* and *in vivo* and have changed the conclusions of the manuscript to state that both CDK1/CycB and CDK2/CycA phosphorylated DENR on Ser73.

In particular, as requested here, we tested by *in vitro* kinase assay a comprehensive panel of CDKs (CDK1/CycB, CDK2/CycA, CDK2/CycE, CDK4/CycD and CDK6/CycD) and the data are presented in Figure 2a-b.

Please note that in the original submission we had not tested CDK1/CycB *in vitro* because of the negative result mentioned above using CDK1 inhibitor *in vivo* on an asynchronous population of cells. Our more careful analysis now, however, adding CDK1 inhibitor specifically to cells in mitosis, shows an effect (Figure 2c-d).

We have also added citations to the papers mentioned by the reviewer.

3) The authors should provide clear references or experimental data (IC50 or Ki values) that seliciclib or A-674563 are exclusively CDK2 inhibitors and that inhibition of CDK1 is not the case *in vivo*, even partially. Alternatively, again, we have no proper answer to what kinase complex are we talking about. Even a minor effect on CDK1 can seriously change the interpretation of the results and the model proposed in the study.

We now provide references showing that we are using seliciclib or A-674563 in a specific range. While seliciclib inhibits CDK1/Cyclin B in a similar dose range as CDK2/Cyclin A (IC50 *in vitro* 0.65µM or 0.7µM respectively, PMID: 9030781), A-674563 does not inhibit CDK1 at all - in fact, it does not affect activity of any other CDK other than CDK2 (PMID: 15956255). Therefore, while there might be some contribution of CDK1 inhibition to the seliciclib effects (Figure 2e,f, Suppl. Figure 2c-g, Suppl. Figure 4e), the effect of A-674563 on DENR Ser73 phosphorylation is free from this potential confounder (Suppl. Figure 2c,g).

Since seliciclib and A-674563 inhibit Erk at an IC50 that is 6 to 20-times higher than the CDK2 IC50, we also tested whether Erk inhibition affects DENR phosphorylation on Ser73, but this is not the case (Suppl. Figure 2h).

That said, as mentioned above, our new data indicate CDK1/CycB also phosphorylates DENR, so we have changed the conclusions of the manuscript regarding the identity of the cell-cycle dependent kinase(s) responsible for this phosphorylation.

4) The effect in Figure 3e is based on very faint bands, but the driven conclusions are strong. The authors should quantify the bands from repeated replica experiments to make a stronger case.

We have both performed and quantified replicates, as well as softened the conclusions from this experiment.

It is not so easy to stabilize DENR protein lacking phosphorylation on Ser73 in mitosis. We tested inhibition of the proteasome with MG132, and this has no effect. We have now included a second replicate of the experiment originally in Figure 3e (now Suppl. Figure 3c) and quantified it (Suppl. Figure 3d). This shows there is indeed a partial rescue of protein levels with the caspase inhibitor Z-VAD-FMK. We have now also introduced a point mutation into the putative caspase cleavage site (D26E) and this also partially rescues protein stability (Figure 3d, Suppl. Figure 3b) as well as the mitotic defects (Figure 5c-d). The fact that the rescue of protein stability is only partial suggests there may be additional mechanisms leading to degradation of non-phosphorylated DENR in mitotic cells, which we now clearly state in the manuscript.

That said, the main question we want to address is whether Ser73 phosphorylation primarily affects protein stability, and as a consequence MCTS1 binding, or the other way around. We think the answer to this is quite clear, that it mainly affects stability and not MCTS1 binding: If we quantify the amount of MCTS1 binding to DENR in a colP and normalize it to the amount of DENR in the IP, if anything, the DENR^{S73A} mutant binds MCTS1 more strongly than wildtype DENR (quantification in Figure 3d) as does the version with the mutated caspase site DENR^{D26E,S73A}. Likewise, in Suppl. Figure 3c, one can see that there is a corresponding increase in MCTS1 colP-ing with DENR^{S73A} when the cells are treated with Z-VAD-FMK (lane 6 vs 3, Suppl. Figure 3c). This would not be the case if the S73A mutation primarily affected MCTS1 binding. This is in agreement with the fact that DENR^{S73A} binds MCTS1 well in asynchronous cells despite not being phosphorylated (Figure 3a) and that we see wildtype DENR binding MCTS1 strongly when expressed in bacteria, where it is not phosphorylated (PMID 29889857).

5) Figure 3f,g seems to contradict Figure 1a-c concerning the ratio of phosphorylated and non-phosphorylated form.

We are not sure we understand the reviewer's concern. In Fig 1a-c, one sees that DENR phosphorylation is highest in mitosis and then drops as the cells exit mitosis. In Figure 3f-g (now Figure 3e), this is only one mitotic timepoint and one timepoint 2 hours later after mitotic exit, where pDENR drops. Please note that the quantification in Figure 3g (now Figure 3e) is a relative quantification, where we normalize the mitotic bar to 1. We are not claiming that all of the DENR is phosphorylated in mitosis, neither in Figure 1 nor in Figure 3. Unfortunately, we cannot confidently conclude the fraction of DENR that is phosphorylated in mitosis because this would require relative quantification of the signal from two different antibodies, which is not possible. However, in Suppl. Figure 6a it looks like the total DENR band shifts quantitatively up in mitotic cells, suggesting it might be almost completely phosphorylated. In Figure 3f (now Figure 3e), the phospho-DENR band looks weaker than in Figure 1 simply because it is a shorter exposure of the blot. One other possible source of misunderstanding is that in Figure 3f (now Figure 3e) there is both endogenous and overexpressed FLAG-tagged DENR, which runs a bit higher, so in the phospho-DENR blot two bands are visible. We have now labeled them explicitly.

6) The authors could discuss the effect of MCTS1[T117A,S118A] mutations in relation with DENR S73A mutation. Is the double site constantly phosphorylated or oscillates in a cell cycle-dependent manner as well?

Unfortunately, despite trying several times, we were not able to generate a functioning antibody that detects phosphorylation on T117, or S118, or both, so we do not know if phosphorylation of these sites is also cell cycle dependent.

We have now tested the effect of the T117A and S118A mutations on protein stability. We find that T117A reduces protein stability (both by itself and in combination with the S118A mutation) whereas S118 mutation alone increases protein stability (Suppl. Figure 1g-h). We have now added this to the results section.

7) The authors raised antibodies against „six predicted phospho-sites in DENR and MCTS1“. However, it is not explained how these sites were predicted. Similarly, on Page 7:“ Hence we mutated to alanine all serines and threonines in DENR and MCTS1 that were observed or predicted to be phosphorylated...“. Please define better how they were observed or predicted and provide a list (some short table that would be easier to grasp than the Excel files in the Supplementary section).

These six sites were observed to be phosphorylated by mass spectrometry and reported at www.phosphosite.org. We have rephrased the text to make this clear.

8) The authors mention that only the DENR Set73 phospho-antibody was generated successfully. But what about the others? Was this the reason why other candidate sites were excluded from the further study?

Yes. We do not know if this was for technical reasons (ie our other antibodies were not good) or for biological reasons (we didn't test the correct cell line or environmental stimulus). We now make this more clear in the text.

9) The authors mention briefly that mass spectrometry was performed on DENR-MCTS1 complex immunopurified from untreated HeLa cells and observed phosphorylation of S73 and T69. The latter site was marked „not shown“. It is not clear why T69 was not studied further, or at least explain why they were not interested in it. Was it quantitative mass spectrometry and were these two sites the only sites that were phosphorylated at the complex? What about the previously mentioned MCTS1-T117,S118?

We did not study T69 further because we were not able to raise a phospho-antibody that detects it. This strongly precludes the type of in-depth analysis we did here for Ser73 (e.g. identifying upstream kinases and physiological regulation). We now make this more explicit in the manuscript.

10) PLK kinases may use CDK phosphorylation sites as primers. The authors could discuss this possibility, especially that they use a substrate complex produced using the bacterial system for in vitro kinase assay (Suppl Figure 2c) that would lack any priming phosphorylations.

This is a good point. We have added a note to this effect in the results section, stating that we may have missed PLK sites because of what the reviewer mentions. (Please note that, as suggested by Reviewer 2, this entire section of the manuscript was shifted to Supplementary Materials.)

Minor points

1) Page 7, please explain in the main text what the „activity ratio >3-fold autophosphorylation“ exactly means: „...identified 50 kinases with the capacity to phosphorylate DENR•MCTS1 in vitro at an activity ratio >3-fold autophosphorylation...“

We have reworded the explanation as follows:

“...identified 50 kinases with the capacity to phosphorylate DENR•MCTS1 in vitro, yielding a signal >3-fold above the background signal caused by kinase autophosphorylation ...”

We hope this makes it more clear.

2) Page 7, wording too bold: just knocking down kinase expression does not provide such a validation ...“We further validated these hits by testing whether any of these kinases regulate DENR•MCTS1 activity in vivo by knocking down their expression and assaying stuORF reporter activity in two cell lines.

We have rephrased it to “We further tested whether any of these kinases regulates DENR•MCTS1 activity...”

3) Page 44 Suppl Figure 2c legend: „DENR•MCTS1 complex or one of its mutants were purified from bacteria“. This sounds like they are bacterial proteins. „Bacterial expression system“ would be more precise.

Done.

4) The sentence: „We exposed cells to different stresses and environmental conditions, including tunicamycin to induce ER stress, high vs. low cell density, staurosporin to induce apoptosis, doxorubicin to induce DNA damage and amino acid and glucose starvation, and in this way discovered that DENR is phosphorylated at Ser73 in a cell cycle-dependent manner (Figure 1):... is unclear, as the results of the experiments performed in all these different conditions are not shown in Figure 1, and it seems like these listed conditions somehow led to the knowledge that Ser73 was phosphorylated in cell cycle-dependent manner and that the following double thymidine block-release experiment was done just to further confirm it.

We have rephrased this sentence to read

“We screened different stresses (ER stress, apoptosis, DNA damage, amino acid and glucose starvation) and environmental conditions (different densities, different cell cycle phases), and discovered that DENR is phosphorylated at Ser73 in a cell cycle-dependent manner”

Reviewer #2

In their manuscript entitled "Cyclin A/CDK2 phosphorylates DENR to orchestrate mitotic protein translation and faithful cell division", Clemm von Hohenberg et al. expand on findings reported in several previous studies from the same lab, which have established the non-canonical translation initiation factor DENR-MCTS1 as a regulator of mRNA translation for specific subsets of transcripts that contain regulatory upstream open reading frames (uORFs). On these transcripts, DENR-MCTS1 mediates re-initiation after uORF translation. Previous work by the Teleman lab and others was dedicated mainly to the identification of (i) DENR-MCTS1 client transcripts, (ii) of the functions and pathways that these transcripts are associated with, and (iii) of the characteristics of the responsible uORFs. In particular, these previous studies established a link between DENR-MCTS1/uORF re-initiation and tissue growth, as well as cancer. Several studies have established a pro-cancer, oncogenic role of DENR-MCTS1.

The current manuscript now addresses a novel aspect of DENR-mediated gene expression control, which is the post-translational regulation of DENR itself. The authors use their findings to present a model according to which DENR is phosphorylated through CDK2/CycA at Serine 73 during the cell cycle. This phosphorylation is observable during M-phase (deposited at mitotic onset; removed at exit), where it is necessary to protect DENR from caspase-mediated proteolytic degradation. Indeed, reporter assays based on several previously identified, cell cycle-relevant DENR targets show preferential DENR regulation in mitotic cells as compared to cells grown under asynchronous conditions. Finally, in DENR knock-out cells (or in a DENR S73A mutant), there is high prevalence of anaphase catastrophe and cell death, presumably due to the lack of translation of the relevant mRNAs.

The presented data are intriguing and novel, of generally good quality, and on a high-interest topic; the described links with the cell cycle could further rationalise some of the previously reported links between DENR and cancer. I therefore encourage the authors to improve the manuscript - as described in two main points and additional other points, below - to bring it in shape for a publication.

We thank the reviewer for the positive assessment and helpful comments for improvement.

In my view, the main weakness of the paper is that it currently does not offer an answer or plausible explanation to the question why this intricate regulation of DENR in the cycle is needed, i.e. why can DENR apparently be degraded cell cycle-dependently,

just to then get “rescued” from the degradation through CDK2-dependent phosphorylation, making DENR abundance finally more-or-less constant over the day? Is this rescue occurring constitutively? Or is the cell cycle-dependent degradation actually sometimes occurring on the WT protein, and under which conditions and with what consequences for the translome?

We believe there are two possible types of reasons for why the mechanism we are describing here exists:

1. Since mitosis is such a short cell cycle stage, regulation needs to occur quickly in order to change protein levels, which can only work post-translationally. Indeed, we see that this mechanism enables cells to quickly degrade DENR when they exit mitosis within 2 hours (Figure 3e).

2. Another option has an evolutionary explanation. There are many examples in evolution where a biological process evolves (e.g. the degradation of DENR in mitosis), and then later instead of removing it, evolution counteracts this by building on it further (ie DENR phosphorylation to block the degradation). One example of this are the genitalia of male *Drosophila*. Most male insect species have genitalia that perform a 180 degree rotation during development, so that the anterior ends up pointing to the posterior. However, for whatever reason, *Drosophila* males need the anterior of their genitalia pointing anterior. So one would think that the easiest solution is to avoid the 180 degree rotation. Instead, *Drosophila* genitalia perform a 180 degree rotation, and then evolution added on an additional 180 degree rotation, for a total 360 degree rotation, returning the anterior to point anterior. This seems like a useless exercise, but it's simply because evolution chose a convoluted path that rational design would not choose. Likewise, the stabilization of DENR during mitosis via a phosphorylation, rather than by removing the mitotic destabilizing element, may simply be evolution's convoluted answer to the problem.

Another conceptually important, open question (related to the previous one) is whether the functional importance of phosphorylation lies in stabilising DENR, or whether it also reprograms the specificity of re-initiation to specific substrates in mitosis as compared to interphase? I believe that with the data in Fig. 4, the authors want to make this claim, but in reality one would need comprehensive evaluation of translation rates (ribosome profiling) to be sure. I think this is the one and only, main experiment that would solve many of my questions, and I am therefore suggesting it below as Main point 1.

The reviewer raises an important point. We have now performed ribosome profiling on DENR^{WT} and DENR^{KO} cells that were either in interphase or enriched for mitosis, to identify the ‘DENR target’ mRNAs during these two phases of the cell cycle, when

Ser73 is differentially phosphorylated. We decided not to synchronize cells in mitosis pharmacologically as others have done, since this stresses the cells and causes strong perturbations of the translome. Instead, we did this by S phase synchronization (double thymidine block) and release into mitosis. It seems to have worked well because we see clear differences in both the RNA-seq profiles and the translome of mitotic cells compared to interphase cells (Suppl. Figure 5a-b). The mRNAs that have higher levels in mitosis have GO enrichment for the cell cycle (adjusted $p = 10^{-17}$) and for changes in the cytoskeleton (adjusted $p < 10^{-3}$) and adhesion (adjusted $p = 10^{-17}$), as expected for mitotic cells (not shown). Comparison of the translation efficiency (TE) of mRNAs in interphase DENR^{KO} vs DENR^{WT} cells (Suppl. Figure 5c-d) recapitulates the set of DENR target mRNAs which we have previously reported (PMID 32938922). So technically the experiment seems to have worked well, yielding a rich and useful dataset which we now provide with this manuscript.

We find that there is generally a good correlation between the DENR targets in interphase cells and in mitotic cells (Suppl. Figure 5e). This suggests that DENR regulates the same set of genes in mitosis as it does in interphase. There are roughly 6 genes such as IL11 which are strongly DENR dependent in interphase cells ($\log_2FC(TE) = -4.4$) but hardly DENR dependent in mitotic cells ($\log_2FC(TE) = -0.2$). These genes, however, are simply not translated in mitosis. IL11 is the single most down-translated gene transcriptome-wide in mitotic wildtype cells. Hence, since it is not well translated in mitosis, it doesn't matter if DENR is present or not. Overall, it does not seem that there are two distinct sets of DENR targets in mitotic and interphase cells. Also consistent with this is the fact that the mitotic increase in translation of the DUSP4 and CDKL5 5'UTR reporters is gone when we mutate the ATGs of the uORFs (Suppl. Figure 4f), indicating that DENR is acting on these mRNAs via the 'usual' mechanism that we have described before in interphase. Hence we conclude that the main functional consequence of DENR Ser73 phosphorylation is on its stability.

The question remains open why our synthetic stuORF reporter does not increase in translation in mitosis, unlike the other reporters carrying the 5'UTRs of endogenous DENR targets. We don't know why. We think this is something technical that is specific to our synthetic stuORF reporter. It is possible that it is not well translated during mitosis for some technical reason.

Main points:

(1) Does the phosphorylation only act through the regulation of DENR abundance, or is there also an aspect of reprogramming of specificity involved? In Figure 4, many questions remain open. Are the reporters mitotically DENR-reinitiated because they are "special" in something (uORFs?), or because there is something

cell cycle-specific about the expression of their RNAs (e.g. RNA abundance regulated across cell cycle, possibly even through uORF and NMD that would also be active on reporter mRNAs) ? The reporter assays as they are currently shown do not give much information on these possibilities; e.g. there are no control reporters with mutated uORFs; there are also no statistics (p-values) shown in Fig. 4B-C.

Please see above the response to the reviewer's general point.

We now have mutated the uORFs in the DUSP4 and CDKL5 reporters and this does indeed blunt their increase in translation caused by mitosis (Suppl. Figure 4f).

From our ribosome profiling experiment, there does not seem to be anything 'special' about the DENR targets in mitosis – they correlate very well with the DENR targets in interphase (Suppl. Figure 5e). The magnitude of the DENR dependence is affected by whether the mRNA is well translated in mitosis or not. The mRNAs that are more dependent on DENR in mitosis than in interphase are the ones that are translationally up-regulated in mitosis (ie their TE in wildtype mitotic cells > TE in wildtype interphase cells) whereas the ones that are less dependent on DENR in mitosis are simply less translated overall in mitosis. (We do not see a correlation to mRNA levels in mitosis).

As requested, we have added statistics to Figures 4b-c.

Suggestion: Why not do ribo-seq on asynchronous and mitotic cells, expressing WT DENR vs. KO (or vs. DENR S73A) ? I would even recommend also including the S73D mutant (phospho-mimic). I am convinced that such an experiment would solve many of the open questions of the study. It would establish whether endogenous DENR client mRNAs are regulated depending on phospho-status and depending on mitotic phase.

We thank the reviewer for this suggestion. The results from this experiment are described above.

(2) Another experiment that should distinguish between functional reprogramming vs. simple abundance effects, could be a mutant in which S73A is combined with a mutation in the caspase site (i.e., mentioned in the manuscript to be at around amino acid 26 – according to the Steitz lab structure, it should be possible to point-mutate this area without affecting interaction with MCTS1 or the ribosome). This should rescue the low stability of S73A alone. Can the authors make and test such a mutant in some of their assays – e.g. stability (like Fig. 3C-D) and functional effects (e.g. assays as in Fig. 5)?

We have now done this. The D26E mutation of the caspase site only partially rescues the stability of the DENR^{S73A} mutant (Figure 3d) and correspondingly it partially rescues the mitotic defects of DENR^{KO} cells (Figure 5c-d, where the D26E,S73A double mutant reduces by half the increased number of cells with mitotic failure compared to control cells). Together with the ribosome profiling experiment mentioned above, these data suggest that the main functional role of S73 phosphorylation is in controlling DENR stability and not in functional reprogramming.

Other Points:

(3) Figure 2: In panel A, the CDK2-Cyclin A2 vs. CDK2/Cyclin E1 effect is beautiful and shows extraordinary specificity - rare to see this so clearly in an *in vitro* assay! I am a little more concerned about the *in vivo* data and the specificity of the drugs at the concentrations used. For example, A-674563 inhibits many kinases in the nM range (see e.g. <https://www.selleckchem.com/products/a-674563.html>: Akt1, PKA, GSK3b, ERK2 all inhibited). For Seliciclib, it seems to be a bit better, but still - the 10 μ M concentration is high (close to the IC50 of, for example, ERK2, 14 μ M). At lower concentrations, well-above the IC50 for CDK2, the effect on pDENR is not impressive (1 μ M Seliciclib; 0.2 μ M A-674563; Figure 2B). Can the authors comment on the specificity of the drugs?

We have now added a significant amount of new data to address this comment, as well as comments by Reviewer 1 asking whether our data can distinguish between CDK2 and CDK1.

Generally, although the *in vitro* IC50 for these CDK2 inhibitors is in the nM range, the concentration that needs to be given to cells to inhibit the kinase is much higher. This is not unusual, and has to do with cellular uptake, efflux and degradation of the drug. Seliciclib is generally used on cells in the μ M range, up to 100 μ M (PMID 15827128, 25378060). To determine the *in vivo* efficacy of Seliciclib, HeLa cells were exposed to the drug for 72 hours and the IC50 towards cell viability was determined at around 14 μ M (PMID 23691435). Hence, application of 1-10 μ M for a duration of only 1-4 hours is in fact a rather low dose *in vivo*.

For A-674563 the IC50 towards CDK2 has not been determined *in vivo*. However its IC50 towards Akt (*in vitro* about 4.2-fold more potency than towards CDK2) was found to be around 100nM and its EC50 at 400nM in MiaPACA2 cells (PMID 15956255). Therefore, doses of 100-1000nM to inhibit CDK2 (as used here) are within this range.

Since the CDK2 inhibitors we used can also inhibit Erk with an IC50 that is 6- to 20-fold higher than the IC50 for CDK2, we also tested whether Erk inhibition affects Ser73 phosphorylation, but this is not the case (Suppl. Figure 2h).

Finally, and importantly, we had previously ruled out that Ser73 is phosphorylated by CDK1 by simply adding a CDK1 inhibitor to asynchronous cells and detecting Ser73 phosphorylation by immunoblotting. Since this had shown no effect (despite blocking histone H3 phosphorylation as a positive control), we had excluded CDK1 as a candidate. However, prodded by the comments of Reviewer 1 we now revisited this conclusion. We now show in Figure 2a an *in vitro* kinase assay with CDK1, 2, 4 and 6 on recombinant DENR•MCTS1 protein, detected with pDENR(S73) antibody, and find that CDK1 phosphorylates Ser73 even more strongly than CDK2. To retest whether CDK1 phosphorylates Ser73 *in vivo*, we used a more careful approach whereby we first synchronized cells at the G2/M transition with a CDK1 inhibitor, then released this block, allowing cells to enter mitosis (e.g. causing an increase in pDENR, lane 3 in Figure 2c), and then briefly re-applied the CDK1 inhibitor RO3306 (Figure 2c-d). This allowed us to test the CDK1 inhibitor on mitotic cells, and we find that it indeed reduces Ser73 phosphorylation. Hence, in sum, we now think that DENR Ser73 is phosphorylated by both CDK1/CycB and CDK2/CycA in mitosis. The relative contribution of these two kinases to Ser73 seems to depend on cell line, with CDK1 predominating in HeLa cells (Figure 2c-d and Suppl. Figure 2g) and CDK2/CycA predominating in U2OS cells (Figure 2e-f and Suppl. Figure 2b).

(4) Do the authors have any information on the approximate fraction of DENR that is phosphorylated in mitosis?

Unfortunately this is difficult to address because it would require the comparative quantification of signal from two different antibodies (the total DENR and phospho-DENR). However, in Suppl. Figure 6a, it looks like total DENR is entirely shifting up in mitotic cells to a slower migrating form on the SDS-PAGE gel. This suggests almost all of DENR protein gets phosphorylated, however we do not know if this is only due to S73. We have added a comment to this effect to the Results section.

(5) Figure 3: In panel A and B: The input image for DENR shows in EV cells high levels of endogenous DENR. Hence, I assume these experiments were not done in DENR knockout cells (it's not described in the legend) - however, wouldn't that have been a better genetic background for these experiments?

Indeed, this is done in DENR^{WT} cells. We tried hard to get DENR^{KO} U2OS cells by CRISPR/Cas9, but were unsuccessful. (They don't like growing as single cells?) Hence all the experiments done in this manuscript with U2OS are done with DENR siRNAs, whereas the experiments done with HeLa use the DENR knockouts. For Figure 3a we used asynchronous HeLa cells, and for Figure 3b we used mitotically

synchronized U2OS cells because they synchronize in mitosis better than HeLa. Since we're IPing the tagged protein, however, it is not necessary to do it in KO cells.

(6) Figure 3C/D: Western blots are notoriously difficult to quantify, as the signal does not scale linearly with protein amount. Comparing/evaluating the wt and mutant time course as it is shown in the western blot is difficult.

Could the authors show an additional, longer Western exposure for the mutant, in which the "0" time-point is similarly strong as the 0 time-point of the WT ? That would make it easier to evaluate if the quantification in panel D corresponds to what one sees in C.

We have added a shorter exposure of the film where the WT protein has a band strength similar to the mutant protein on the longer exposure.

Please note that all our western blots are not detected with film, which has the linearity issues raised by the review, but with a Biorad Chemidoc Imager which has linear detection and a huge dynamic range of >65000 shades of grey.

(7) Figure 3F: I find the DENR rescue by Z-VAD-FMK not very convincing. For a start, more replicates and quantification would be needed. Maybe the best experiment would be the mutation of the caspase site, see above point (2).

We agree that the rescue of DENR protein levels caused by Z-VAD-FMK is not very strong. We have now included a second replicate of the experiment originally in Figure 3e-f (now Suppl. Figure 3c) and quantified it (Suppl. Figure 3d). This shows there is indeed a partial rescue of protein levels with the caspase inhibitor Z-VAD-FMK. As suggested, we have now also introduced a point mutation into the putative caspase cleavage site (D26E) and this also partially rescues protein stability (Figure 3d, Suppl. Figure 3b) as well as the mitotic defects (Figure 5c-d). The fact that the rescue of protein stability is only partial suggests there may be additional mechanisms leading to degradation of non-phosphorylated DENR in mitotic cells, which we now clearly state in the manuscript.

That said, the main question we want to address is whether Ser73 phosphorylation primarily affects protein stability, and as a consequence MCTS1 binding, or the other way around. We think the answer to this is quite clear, that it mainly affects stability and not MCTS1 binding: If we quantify the amount of MCTS1 binding to DENR in a colP and normalize it to the amount of DENR in the IP, if anything, the DENR^{S73A} mutant binds MCTS1 more strongly than wildtype DENR (quantification in Figure 3d)

as does the version with the mutated caspase site DENR^{D26E,S73A}. Likewise, in Suppl. Figure 3c, one can see that there is a corresponding increase in MCTS1 colP-ing with DENR^{S73A} when the cells are treated with Z-VAD-FMK (lane 6 vs 3, Suppl. Figure 3c). This would not be the case if the S73A mutation primarily affected MCTS1 binding. This is in agreement with the fact that DENR^{S73A} binds MCTS1 well in asynchronous cells despite not being phosphorylated (Figure 3a) and that we see wildtype DENR binding MCTS1 strongly when expressed in bacteria, where it is also not phosphorylated (PMID 29889857).

(8) Figure 5D: statistics should be added. Do the differences pass the typical tests for statistical significance?

Done. In order to perform statistical tests, we now group and show the data slightly differently (Figure 5c-d). This shows that the differences are statistically significant.

(9) Finally: The structure of the manuscript is a little bit awkward and could be much improved, especially at the beginning. For example, for the entry to the results section, there are a lot of "negative results" (the various phospho-site mutants) that are not of immediate further relevance to the overall narrative – they take a lot of space and distract from the main thrust. Some of this is confusing for the beginning of a paper. Example on page 5: "Worth mentioning is that we performed these experiments in HeLa cells in standard culture conditions, where many of these residues may not be phosphorylated. Hence, in the future it may be worth testing this panel of mutants also in other cell lines or environmental conditions. Furthermore, we used asynchronous cells, explaining in part why the DENR S73A mutant, which is relevant only in mitosis (see below), did not have a strong phenotype." – some of these comments only become clear later in the manuscript. I believe this actually destabilises, rather than helps, the reader here.

Moreover, just after the part cited above, the description how the authors finally settled on studying Ser73 reads very arbitrary – and although this may honestly be how the project in reality developed, I recommend presenting in a slightly more targeted fashion.

Thus, I would suggest restructuring and/or describing some of the less relevant experiments (read: less relevant for the actual story the authors want to tell) much shorter in the main text, and moving a lot of the description/interpretation to the supplementary figure legend or, in part, to discussion. I still think the "negative" data should be reported (it may indeed be helpful for others), but it should take less space and be not so prominently placed.

In general, the introduction and discussion could also profit from a bit better integration and flow in the arguments.
In summary: The data could be much more effectively presented.

As suggested, we have shifted all the data and the corresponding text relating to our testing of various kinases and phospho-sites out of the main text and into Supplementary Materials. We hope the manuscript is now easier to read.

Reviewer #3

Hohenberg et al address whether and how activity of the DENR-MCTS1 complex is regulated by posttranslational modifications. They report a comprehensive analysis of modifications and assess their functional significance in appropriate mutants. They find that DENR undergoes CDK2-dependent phosphorylation on Ser73 in mitosis and suggest that this modification is important to promote translation of specific mitotic target genes and error-free mitoses.

This is an interesting model and a useful contribution to the emerging view that selective translational regulation of proteins involved in mitosis is essential for faithful, correct mitosis. Although the phenomenon of cell-cycle-specific translation has been described by a number of labs, the mechanisms behind it are not well understood.

They have provided compelling evidence that DENR-S73 phosphorylation is cell-cycle dependent and high in mitosis; that DENR is phosphorylated on S73 by CDK2-CycA in vitro and in vivo; and this phosphorylation does not affect the interaction with MCTS1 but affects the stability of DENR in mitosis.

They then explore whether DENR-S73 phosphorylation has an effect on mitotic translation and thereby on mitosis. They use reporter constructs using the 5'UTR-s of selected DENR target genes and show that DENR is required for their optimal translation. They also assess the levels of some of the endogenous proteins with known mitotic functions. Finally, they show that in the absence of DENR cells accumulate in mitosis and have aberrant mitoses.

Overall, the manuscript is very clearly written and the data are consistent with the model. However, I believe some control experiments and data analyses would strengthen the conclusions.

We thank the reviewer for the positive evaluation and the helpful comments ! We have added new data to address the concerns raised by the reviewer, and we believe this has made the manuscript stronger.

1) Fig 1 d : The localization of P-DENR in mitotic cells was investigated. The authors claim that the P-DENR antibody lights up the contractile ring in telophase. A good control would be the KO transformed with DENR-S73A. Notably, the midbody is notorious for staining positive with a number of phospho-specific antibodies. Furthermore, according to their model DENR is degraded in a caspase-dependent manner in anaphase - telophase, making it less likely that the brightly stained signal in telophase is due to DENR. On the other hand, if it is, it supports their suggestion that there is local translation at this site or that DENR may have a function other than translation.

We thank the reviewer for this point. Indeed, it turns out that the midbody staining is not specific, since it does not decrease upon DENR knockdown (Suppl. Figure 1c). We now mention this explicitly in the text and have removed the arrows pointing to the midbody staining in the main figure. (The stainings shown in Fig 1d are on U2OS cells and despite trying several times we never obtained a DENR knockout U2OS line by CRISPR/Cas9, hence the experiment was done with a DENR siRNA.)

2) The putative target proteins were not enriched in mitosis as much in KO as in wt, this phenotype was rescued by transfection with wt DENR – but rescue should be addressed also with the S73A mutant, either using the reporter and/or measuring endogenous protein levels. Specifically, at least some of the experiments in fig 4 and 5 should be repeated in a KO transfected with the S73A mutant. The only phenotypic analysis with the DENR-S73A mutant is shown in fig 5d, but as discussed in point 4-5, the analysis of mitotic progression could be improved.

We thank the reviewer for this suggestion. We now tested the S73A mutant phenotypically in two different ways. We reconstituted the DENR^{KO} cells with either DENR^{WT} or DENR^{S73A} and measured the level of target proteins in mitosis (Figure 4j-k). As expected, this shows that the target proteins are significantly lower in cells reconstituted with the DENR^{S73A} mutant (lane 2) compared to DENR^{WT} (lane 1). We also tested whether the DENR^{S73A} mutant can rescue the mitotic defects caused by DENR loss-of-function (Figure 5c-d). As expected, here too, the DENR^{S73A} mutant does not rescue. We think this is an important finding because it shows for the first time, to our knowledge, that the translation that is occurring during mitosis is important for mitosis to proceed normally (because the DENR^{S73A} mutant is only defective in

phosphorylation, which only occurs in mitosis). It also confirms that the DENR^{S73A} mutant is not functional.

3) U2OS cells were used in figures 1-4, then in Fig 5 HeLa cells were used to show an accumulation of cells in mitosis in DENR^{KO}. Is this phenomenon specific for HeLa cells? Why is the model system changed? In general, it would be reassuring to see some of the key experiments performed in more than one cell line, and perhaps also in a non-transformed cell line.

The reason we have data from two different cell lines is that the HeLa cells do not synchronize their cell cycles as well as U2OS cells. Unfortunately, however, we were not able to obtain DENR^{KO} U2OS cells by CRISRP/Cas9 despite trying several times. (We think this is for technical reasons having to do with the puromycin selection in U2OS cells and the requirement that cells grow as single clones.) Hence we need to do siRNA-mediated DENR knockdowns in the U2OS cells, which stresses them and makes it difficult to get them synchronized properly. Reconstitution experiments requiring two transfections for siRNA-mediated knockdown and plasmid transfection make synchronization impossible in the U2OS. Instead, in the HeLa cells we have DENR^{KO} lines which we can use for reconstitution experiments with mutant DENR versions but they are less good for synchronization.

That said, we have now added additional data so that almost all the key data points are shown in one way or another in both HeLa and U2OS cells:

	HeLa	U2OS
Phospho S73 is high in mitosis	Fig 2c	Fig 1
S73 is phosphorylated by CDK1+2	Fig 2 & S2	Fig 2 & S2
S73 affects DENR mitotic stability	(Fig 4j mildly)	Fig 3
DENR affects target translation in mitosis	Riboseq Fig S5	Lucif reporters Fig 4
S73 affects target translation in mitosis	Fig 4j	(Fig S4e for CDK2 inhibition)
Affects mitosis	Fig 5	Fig S7b

4) Fig 5d shows two biological repeats and assessing the stages of mitosis based on DAPI staining. This is very difficult to do based on DAPI alone and if one is to conclude anything about the problems in mitosis, better images and a better approach are required. See also point 5.

We have now added a 3rd biological replicate and statistical analyses to Figure 5d (now Figure 5c-d) showing that the effects are statistically significant – loss of DENR leads to fewer cells in late stages of mitosis and more cells showing aberrant mitotic DNA conformations. This is rescued by DENR^{WT} but not DENR^{S73A}. We believe the DAPI staining can be used well to stage cells in the various stages of mitosis, and the

mitotic blebs are visually quite obvious. We have now also added as Suppl. Figure 7c spindle staining to show that loss of DENR leads to irregular and multipolar spindles.

5) In the Discussion the authors describe the phenotype upon DENR KO as anaphase catastrophe, largely because of the involvement of CDK2 in DENR phosphorylation. However, anaphase catastrophe is characterized by multipolar spindles rather than “mitotic blebs” and in this work it is not addressed in detail how the cells die, only that they die in mitosis. Therefore it is not correct to state that “DENR loss of function phenocopies this mitotic failure”(ie anaphase catastrophe). Even though the model of anaphase catastrophe is an attractive one, they should demonstrate it by eg live-cell imaging or immunofluorescence and quantification of multipolar spindles. At the very minimum, they should reformulate the statement and describe the phenotype as aberrant mitoses or similar.

We have reformulated the statement as “aberrant mitosis” and “atypical mitotic figures and mitotic blebs”. As suggested by the reviewer, we also tested if DENR mutant cells have multipolar spindles which characterize anaphase catastrophe by staining for γ -tubulin, and indeed this is the case (Suppl. Figure 7c).

6) An overlap between the DENR-dependent transcripts and transcripts whose translation is increased in mitosis would strengthen the model. The datasets available from the literature – and cited in the manuscript (5, 8) – and their datasets on DENR-dependent transcripts should be compared. See also point 7.

We have now performed an extensive RNA-seq and Ribosome Profiling analysis comparing DENR^{WT} to DENR^{KO} cells in interphase and in mitosis (Suppl. Figure 5). From this, we identify:

- 1326 transcripts corresponding to 694 genes with elevated mRNA levels in mitosis (Suppl. Figure 5a, Suppl. Dataset 5) and none with reduced mRNA levels, probably due to the short duration of mitosis.
- 266 transcripts (181 genes) that are translationally up-regulated in mitotic cells and 1090 transcripts (696 genes) that are translationally down-regulated in mitotic cells (Suppl. Figure 5b, Suppl. Dataset 6).
- 1108 transcripts (653 genes) and 990 transcripts (576 genes) as DENR targets in interphase and mitotic cells, respectively (Suppl. Figure 5c-d, Suppl. Dataset 7).

Interestingly, a comparison of the change in translation efficiency upon DENR loss in interphase versus mitotic cells shows a good correlation (Suppl. Figure 5e) suggesting that in general the mRNAs that are DENR targets in mitosis are also DENR targets in

interphase, although to varying degrees. A hand-full of mRNAs appear to be strong DENR targets in interphase cells but not in mitotic cells (Suppl. Figure 5e), but these genes are amongst the ones with the strongest drop in translation efficiency transcriptome-wide in wildtype mitotic cells compared to interphase cells (e.g. IL11). This suggests that these mRNAs are not well translated in mitosis, and hence are not sensitive to DENR loss.

We looked for overlap between our datasets and those in the literature, but there is little overlap (also comparing the different published datasets to each other). We think this has to do with the method of synchronization. Drugs used by the other studies to synchronize cells, such as nocodazole, and to a lesser extent the Eg5 inhibitor STLC, induce some cell stress and hence perturb the translome (PMID 29963178, 26824457). Instead, we used S phase synchronization and release which doesn't perturb the translome, but perhaps shows a less stringent mitotic enrichment (due to desynchronization during the hours of release) and might therefore lead to underestimation of the effect size.

Nonetheless, the main point of the reviewer is that we should test the overlap between DENR-dependent transcripts and those whose translation efficiency increases in mitosis. Since we did both RNA-seq and Riboseq, we can address this question using these new datasets we generated. Interestingly, of the 266 transcripts that are translationally up-regulated during mitosis in wildtype cells, 114 of them are DENR targets. This fraction (~40%) is significantly higher than the 10% of transcripts that are DENR targets in interphase cells ($p=0$ – ie less than the computer can calculate - by binomial distribution). Hence DENR appears to play a particularly important role in mitotic translation.

7) The finding that the synthetic stuORF-reporter does not show DENR-dependent increased translation in mitosis is rather puzzling, and it would be interesting to have the authors' view on the possible reason. They even report an increase of the translation in mitosis of DENR-dependent genes that have not been associated with mitosis (Fig 4), but not of the synthetic reporter. Does the synthetic stuORF lack something that the 5'UTR-s of the mitotic genes have? Do they have anything in common other than the presence of small uORF-s? Or is it that even though DENR phosphorylation and stabilization correlate with mitotic translation, it is not sufficient for the mitotic DENR-targets being efficiently translated? This issue brings us back to the importance of performing the rescue experiments with the DENR-S73A mutant suggested in point 2.

We agree with the reviewer – these are interesting points. We were also puzzled by the fact that the synthetic stuORF reporter behaves as an outlier and does not increase

its translation in mitosis like the other reporters carrying the 5'UTRs of endogenous DENR targets. We have now looked at this in more detail:

One possibility was that there is something particular about the mRNAs that are DENR targets in mitosis compared to the mRNAs that are DENR targets in interphase. For this reason, we performed the ribosome profiling described above. We find, however, that the mRNAs that are DENR targets during interphase are also targets in mitosis, and the other way around (Suppl. Figure 5e). There are roughly 6 genes such as IL11 which are strongly DENR dependent in interphase cells ($\log_2\text{FC}(\text{TE}) = -4.4$) but hardly DENR dependent in mitotic cells ($\log_2\text{FC}(\text{TE}) = -0.2$). These genes, however, are simply not translated in mitosis. IL11 is the single most down-translated gene transcriptome-wide in mitotic wildtype cells. Hence, since it is not well translated in mitosis, it does not matter if DENR is present or not. Conversely, the mRNAs that are more dependent on DENR in mitosis than in interphase are the ones that are translationally up-regulated in mitosis (ie their TE in wildtype mitotic cells $>$ TE in wildtype interphase cells). So we think the set of DENR targets in mitosis is basically the same as the DENR targets in interphase. This fits with the fact that also the reporters for the DENR targets we previously identified in interphase cells, which have no known function specifically in mitosis, increase in mitotic cells (Figure 4c). Also consistent with this is the fact that the mitotic increase in translation of the DUSP4 and CDKL5 5'UTR reporters is gone when we mutate the ATGs of the uORFs (Suppl. Figure 4f), indicating that DENR is acting on these mRNAs via the 'usual' mechanism that we have described before in interphase. These data indicate that Ser73 phosphorylation does not affect DENR function beyond its role in stabilizing the protein.

To test whether there is something special about the uORFs in the CDKL5 5'UTR, we cloned them into a synthetic background (Reviewer Figure 1 below). The stuORF reporter is made by inserting a stuORF into the 5'UTR of Lamin B1. Therefore, we cloned either CDKL5 uORFs 1 + 2 together (which are overlapping), or uORF3 into the Lamin B1 5'UTR. These reporters, however, do not increase in translation in mitosis, similar to the stuORF reporter, suggesting there is nothing special about these uORFs.

Reviewer Figure 1: The CDKL5 uORFs do not cause increased translation in mitosis when cloned into the LaminB1 5'UTR.

In sum, our best guess is that the synthetic stuORF reporter is lacking something that it would need to be translated in mitosis (analogous to IL11 mentioned above). We think this is an interesting topic which is worth studying in the future.

Regarding the DENR-S73A mutant, please see our response to point #2 above.

Minor comments:

1) In the Methods section the nocodazole concentration is described as 400 ng/ml. This seems very high, even 40 ng/ml is sufficient to arrest the cells at G2/M. Although this is unlikely to change the conclusions, such a high concentration would be a severe stress to the cells, leading to some cells arresting in interphase, and this should be considered when interpreting the results. Unless it was just a typo.

We are indeed using 400ng/mL. This dose is within the range people use (PMID: 27317434) - even up to 1.2 µg/mL have been used for U2OS cells which require higher doses than, eg HeLa cells (PMID:18792104). We would like to mention that presynchronization with thymidine (S phase), followed by release into nocodazole clearly reduces toxicity, because – as opposed to most nocodazole toxicity studies – cells are not in mitotic arrest for prolonged periods of time, since gradual mitotic enrichment only starts about 10 hours after release, 3 hours before analysis (Figure 1a, pHH3 blot).

That said, we agree with the reviewer that synchronizing cells with nocodazole (also at lower concentrations) causes stress, which is why we're also testing our hypotheses in unsynchronized cells (e.g. Fig 1d, 4h-i, 5a-d). Additionally, we have confirmed our results in cells synchronized by the Eg5 inhibitor STLC (Figure 3d, Figure 4j,k, Suppl. Figure 4f, Suppl. Figure 6a,d,f) or after release from an S phase block without mitotically synchronizing agent (RiboSeq, Suppl. Figure 5).

2) "U2OS or HeLa cells were plated in 15cm dishes on siRNA and Lipofectamine RNAiMAX transfection reagent" - something must be missing here.

Thanks for catching that – we have rephrased it to "U2OS or HeLa cells were reverse transfected in 15cm dishes containing siRNA and Lipofectamine RNAiMAX transfection reagent..."

REVIEWERS' COMMENTS

Reviewer #1 (Remarks to the Author):

The authors have considerably improved the manuscript and added new experiments. They have followed my advice and found that cyclin B is indeed involved in the described phosphorylation mechanism as well. The authors have therefore reached a new conclusion and have changed the title, the abstract, and the text accordingly. I recommend publishing after a minor change of the text as explained below.

Concerning the cyclin specificity, I would recommend adding another sentence (or sentences) to explain the nice results that they have presented in Fig 2a,b. This result is in good agreement with the gradually increasing substrate specificity of cyclin CDK complexes (first discovered in ref 46,47 and later confirmed in mammalian cells ref 49). Unfortunately, this important mechanism, which is conserved in CDK regulation of eukaryotes has been very often missed, and sadly, this was also the reason why the current study initially failed to provide a full and meaningful picture of the roles of different CDK complexes in the described mechanism. It is very important for the whole cell cycle and cancer field to properly communicate this conserved principle - a gradual increase of intrinsic activity in the order of appearance of the cyclin in the cell cycle. This would avoid many mistakes and erroneous hypotheses and unnecessary work in the future: later CDK complexes are always more specific towards the consensus CDK phosphorylation sites. The current work and Fig 2b provide yet another proof of this conserved principle. Cyclins are not merely the activators of CDK, but they also gradually modulate the intrinsic active site-specificity of CDK.

Reviewer #2 (Remarks to the Author):

The authors have gone to great efforts to answer all my points in a satisfactory fashion and to restructure the manuscript.

I am happy to see that this interesting story has improved so significantly, both in terms of data and presentation.

Reviewer #3 (Remarks to the Author):

The revised version of the manuscript is considerably improved. They have provided an impressive amount of additional data that strengthens the paper and is a gold mine for researchers working on cell-cycle-regulated translation. They have also performed additional control experiments, including technical controls (eg for IF staining), rescue experiments and the major experiments are now performed in more than one cell line. They have satisfactorily addressed all my concerns and comments.

I have one minor suggestion on how to formulate an important point.

As they point out in the rebuttal letter, it is an important finding that the translation occurring during mitosis is important for normal mitoses. On p16 of the manuscript they say "Since DENRKO cells reconstituted with DENRS73A have mitotic defects, and since Ser73 is specifically phosphorylated in mitosis, these data show that the translation occurring during mitosis is important functionally for mitosis to proceed correctly." This statement is lacking the importance of DENR for mitotic translation, even if it is clear from other parts of the paper. Elsewhere they do mention the number of transcripts whose translation is dependent on DENR (p 12), but only in the rebuttal letter do they point out that of the 266 transcripts that are translationally up-regulated during mitosis, 114 are DENR targets and that this fraction (~40%) is significantly higher than the 10% of transcripts that are DENR targets in interphase cells.

In my mind these numbers and statements belong directly together, to make the conclusion even clearer. Having said that, all the information is there in the manuscript.

Reviewer #1

The authors have considerably improved the manuscript and added new experiments. They have followed my advice and found that cyclin B is indeed involved in the described phosphorylation mechanism as well. The authors have therefore reached a new conclusion and have changed the title, the abstract, and the text accordingly. I recommend publishing after a minor change of the text as explained below.

Concerning the cyclin specificity, I would recommend adding another sentence (or sentences) to explain the nice results that they have presented in Fig 2a,b. This result is in good agreement with the gradually increasing substrate specificity of cyclin CDK complexes (first discovered in ref 46,47 and later confirmed in mammalian cells ref 49). Unfortunately, this important mechanism, which is conserved in CDK regulation of eukaryotes has been very often missed, and sadly, this was also the reason why the current study initially failed to provide a full and meaningful picture of the roles of different CDK complexes in the described mechanism. It is very important for the whole cell cycle and cancer field to properly communicate this conserved principle - a gradual increase of intrinsic activity in the order of appearance of the cyclin in the cell cycle. This would avoid many mistakes and erroneous hypotheses and unnecessary work in the future: later CDK complexes are always more specific towards the consensus CDK phosphorylation sites. The current work and Fig 2b provide yet another proof of this conserved principle. Cyclins are not merely the activators of CDK, but they also gradually modulate the intrinsic active site-specificity of CDK.

We thank the reviewer again for these helpful and stimulating comments and explanations. Based on this we have now added the following sentence to the manuscript:

“These results fit with the principle that cyclins modulate Cdk specificity, and that as cells progress through the cell cycle towards mitosis Cyclin/CDK complexes become progressively more specific for the consensus CDK phosphorylation motif ^{45, 46, 47}.”

Reviewer #2

The authors have gone to great efforts to answer all my points in a satisfactory fashion and to restructure the manuscript.

I am happy to see that this interesting story has improved so significantly, both in terms of data and presentation.

We thank the reviewer for his/her positive feedback.

Reviewer #3

The revised version of the manuscript is considerably improved. They have provided an impressive amount of additional data that strengthens the paper and is a gold mine for researchers working on cell-cycle-regulated translation. They have also performed additional control experiments, including technical controls (eg for IF staining), rescue experiments and the major experiments are now performed in more than one cell line. They have satisfactorily addressed all my concerns and comments.

I have one minor suggestion on how to formulate an important point.

As they point out in the rebuttal letter, it is an important finding that the translation occurring during mitosis is important for normal mitoses. On p16 of the manuscript they say "Since DENRKO cells reconstituted with DENRS73A have mitotic defects, and since Ser73 is specifically phosphorylated in mitosis, these data show that the translation occurring during mitosis is important functionally for mitosis to proceed correctly." This statement is lacking the importance of DENR for mitotic translation, even if it is clear from other parts of the paper. Elsewhere they do mention the number of transcripts whose translation is dependent on DENR (p 12), but only in the rebuttal letter do they point out that of the 266 transcripts that are translationally up-regulated during mitosis, 114 are DENR targets and that this fraction (~40%) is significantly higher than the 10% of transcripts that are DENR targets in interphase cells.

In my mind these numbers and statements belong directly together, to make the conclusion even clearer. Having said that, all the information is there in the manuscript.

We thank the reviewer for this comment. As suggested by the reviewer, we have now added the following sentence in the Results section of the manuscript:

"Furthermore, the DENR/MCTS1 complex appears to be playing a particularly important role during mitosis because 114 of the 226 transcripts which are translationally up-regulated during mitosis are DENR targets (ie 40%), which is a significantly larger fraction than the 10% of transcripts that are DENR targets in interphase cells ($p=0$ by binomial distribution)."